# Atypical APC/C-dependent degradation of Mcl-1 provides an apoptotic timer during mitotic arrest

Lindsey A Allan[1,†], Agnieszka Skowyra[1,†], Katie I Rogers[1], Désirée Zeller[1] & Paul R Clarke[1,2,*]

## Abstract

The initiation of apoptosis in response to the disruption of mitosis provides surveillance against chromosome instability. Here, we show that proteolytic destruction of the key regulator Mcl-1 during an extended mitosis requires the anaphase-promoting complex or cyclosome (APC/C) and is independent of another ubiquitin E3 ligase, SCF[Fbw7]. Using live-cell imaging, we show that the loss of Mcl-1 during mitosis is dependent on a D box motif found in other APC/C substrates, while an isoleucine-arginine (IR) C-terminal tail regulates the manner in which Mcl-1 engages with the APC/C, converting Mcl-1 from a Cdc20-dependent and checkpoint-controlled substrate to one that is degraded independently of checkpoint strength. This mechanism ensures a relatively slow but steady rate of Mcl-1 degradation during mitosis and avoids its catastrophic destruction when the mitotic checkpoint is satisfied, providing an apoptotic timer that can distinguish a prolonged mitotic delay from normal mitosis. Importantly, we also show that inhibition of Cdc20 promotes mitotic cell death more effectively than loss of APC/C activity through differential effects on Mcl-1 degradation, providing an improved strategy to kill cancer cells.

**Keywords** apoptosis; mitosis; Mcl-1; mitotic cell death; proteolysis
**Subject Categories** Cell Cycle; Post-translational Modifications, Proteolysis & Proteomics
The EMBO Journal (2018) 37: e96831

## Introduction

Cellular responses to the disruption of mitosis include the induction of cell death by apoptosis, which normally prevents the propagation of chromosomal abnormalities that result from defects in mitotic spindle assembly. Identification of the mechanisms controlling the induction of apoptosis during mitosis is not only important to understand how chromosome instability is generated in cancer, but also how cells respond to anti-cancer chemotherapeutics such as microtubule poisons that target the spindle and cause an arrest or prolonged delay in mitosis.

The timing of exit from mitosis is determined by the mitotic or spindle assembly checkpoint. During prometaphase, kinetochores that are unattached to spindle microtubules signal through the Mps1-dependent generation of the mitotic checkpoint complex (MCC), which inhibits the Cdc20-dependent activation of the anaphase-promoting complex or cyclosome (APC/C), an E3 ligase that ubiquitinates key substrates and targets them for destruction by the proteasome (Pines, 2011; Sivakumar & Gorbsky, 2015). If the proper bi-allelic attachment of microtubules to kinetochores fails or is prevented pharmacologically, for instance by a drug that disrupts microtubule dynamics, then a cell is held in mitosis for a prolonged period by sustained activation of the checkpoint.

A key substrate of APC/C$^{Cdc20}$ is cyclin B1, the regulatory subunit of the CDK1-cyclin B1 protein kinase that acts as the master regulator of mitosis. During a prolonged mitotic arrest, the loss of cyclin B1 is restrained, although it eventually drops below the threshold required to maintain the arrest despite the checkpoint (Brito & Rieder, 2006). The propensity of cells to undergo apoptosis correlates with the duration of mitosis (Bekier et al, 2009; Huang et al, 2009; Colin et al, 2015), which indicates a progressive increase in pro-apoptotic signalling until it reaches a threshold sufficient to initiate cell death. Whether or not a cell dies in mitosis depends upon whether the apoptotic threshold is breached before exit from mitosis (Gascoigne & Taylor, 2008). Entry into interphase is generally associated with increased cell survival, probably due to a raised apoptotic threshold, although cells that have been arrested for a prolonged period in mitosis can subsequently undergo cell cycle arrest or post-mitotic cell death (Bekier et al, 2009; Huang et al, 2009; Uetake & Sluder, 2010; Colin et al, 2015).

Mcl-1 has emerged as a major determinant of the cellular response to microtubule poisons that target mitotic cells (Harley et al, 2010; Shi et al, 2011; Wertz et al, 2011; Dikovskaya et al, 2015; Haschka et al, 2015; Topham et al, 2015; Sloss et al, 2016). Mcl-1 is a member of the anti-apoptotic Bcl-2 family of proteins that suppress the activation of caspases, which are apoptotic proteases that bring about cellular destruction (Budihardjo et al, 1999). Antagonism of anti-apoptotic Bcl-2 family proteins, which are often over-expressed in tumours, has been proposed as a valuable anti-cancer strategy (Opferman, 2016). Importantly, Mcl-1 is degraded by an ubiquitin-proteasome-dependent mechanism in response to the disruption of mitosis (Harley et al, 2010; Wertz et al, 2011).

1   Division of Cancer Research, Jacqui Wood Cancer Centre, School of Medicine, Ninewells Hospital and Medical School, University of Dundee, Dundee, UK
2   The University of Queensland Diamantina Institute, Faculty of Medicine, Translational Research Institute, Woolloongabba, Qld, Australia
    *Corresponding author. Tel: +61 7344 37990; E-mail: paul.clarke@uq.edu.au
    †These authors contributed equally to this work

Removal of Mcl-1 promotes mitotic cell death while stabilisation of the protein inhibits apoptosis induced by mitotic arrest (Colin *et al*, 2015; Topham *et al*, 2015; Sloss *et al*, 2016). In addition, Mcl-1 controls cell fate through suppression of both caspase-dependent activation of DNA damage signalling at telomeres during a prolonged mitosis and the subsequent activation of p53 in cells that initially survive mitotic arrest and enter interphase (Colin *et al*, 2015; Hain *et al*, 2016). Mcl-1 is therefore a key modulator of effects of mitotic disruption, and the mechanisms that determine Mcl-1 instability during mitotic arrest are crucial to understand cellular responses to anti-mitotic drugs.

The ubiquitin E3 ligase that catalyses the poly-ubiquitylation of Mcl-1 and primes it for proteasomal destruction during mitosis is therefore of significant interest. Previous biochemical experiments have suggested roles for SCF[Fbw7] (Wertz *et al*, 2011) and APC/C[Cdc20] (Harley *et al*, 2010) in the control of Mcl-1 stability during mitosis, although Fbw7 might have an indirect role through control of the progression of mitosis (Finkin *et al*, 2008) and the requirement for Cdc20 for mitotic Mcl-1 destruction has been questioned (Díaz-Martínez *et al*, 2014; Sloss *et al*, 2016). In this report, we establish by live-cell imaging that the loss of Mcl-1 during mitotic arrest requires APC/C and not SCF[Fbw7]. We show that Mcl-1 is an atypical APC/C substrate with an unusual dependency on Cdc20. A C-terminal isoleucine-arginine (IR) motif makes recognition of Mcl-1 by APC/C insensitive to the level of Cdc20. This mechanism ensures that Mcl-1 is degraded during a prolonged mitosis at a relatively slow rate that is not controlled by the spindle assembly checkpoint. Mcl-1 destruction is not accelerated when the checkpoint is relieved, preventing catastrophic apoptosis as a cell progresses out of mitosis. We propose that the steady rate of Mcl-1 loss during mitotic arrest enables it to act as a timer that measures the period of mitotic arrest independently of checkpoint strength and thereby distinguishes severe mitotic disruption from the normal progression of mitosis. Consistent with an atypical mechanism of APC/C-mediated Mcl-1 destruction, we show that distinct means of inducing mitotic arrest have differential effects on Mcl-1 degradation compared to that of cyclin B1, yielding alternative cell fate profiles. These properties provide opportunities to enhance cell killing in response to chemotherapeutic drugs that target mitotic cells, particularly in cancer cells that are prone to mitotic slippage. Thus, we show that the propensity to undergo mitotic cell death is determined by both the temporal period of mitosis and the nature of the mitotic arrest.

# Results

## Mcl-1 is targeted for degradation during mitotic arrest by APC/C

Previous work has identified two ubiquitin E3 ligases, namely APC/C and SCF[Fbw7], as determinants of Mcl-1 loss in response to microtubule poisons (Harley *et al*, 2010; Wertz *et al*, 2011). To test the relative contributions of these enzymes to Mcl-1 degradation specifically during mitotic arrest, we analysed by Western blotting the requirement for each E3 ligase in pools of isolated rounded-up mitotic cells synchronised in the period of mitotic arrest by treatment with the microtubule poison nocodazole (830 nM). siRNA-mediated knockdown of APC11, a catalytic subunit of APC/C, stabilised Mcl-1 in HeLa cells during a synchronised mitotic arrest

(Fig 1A). Although knockdown of Fbw7 (encoded by the FBXW7 gene and also known as hCDC4; Davis *et al*, 2014) elevated Mcl-1 protein levels in interphase, it did not prevent Mcl-1 degradation during mitotic arrest (Fig 1A). While Mcl-1 degradation did appear to be slightly reduced in DLD1 FBW7[−/−] knockout cells treated with 100 nM paclitaxel (Fig EV1A and B), the mitotic arrest in these cells was compromised compared with wild-type cells as demonstrated by lower levels of both MPM2 and phosphorylated APC3 and by the reduced number of cells positive for histone-H3 phosphorylated on Ser10 (p-Ser10-H3), consistent with previous data (Finkin *et al*, 2008; Wertz *et al*, 2011). Importantly, when a higher concentration of paclitaxel (500 nM) was used to better maintain mitotic arrest in DLD1 FBW7[−/−] cells (Fig EV1C), Mcl-1 degradation occurred as in wild-type cells (Fig 1B). As observed in HeLa cells, knockdown of APC11 also stabilised Mcl-1 in mitotically arrested DLD1 WT cells compared to control knockdown cells (siLuc) (Fig 1C). These results indicate that Mcl-1 degradation during mitotic arrest is mediated by the APC/C, whereas loss of Fbw7 causes an apparent stabilisation of Mcl-1 because of its ability to promote premature exit from mitosis rather than by inhibiting Mcl-1 degradation during mitotic arrest.

To analyse rates of protein destruction more accurately and exclusively in mitotic cells with precise correlation to the timing of mitosis, we established cell lines expressing inducible YFP-tagged Mcl-1 and used live-cell imaging of the fluorescent fusion protein to monitor events in individual mitotic cells. YFP-Mcl-1 was degraded during mitotic arrest similar to endogenous Mcl-1 (Fig EV1D). However, compared with some other APC/C substrates, such as Nek2A and cyclin A, which are degraded rapidly despite an active checkpoint (Hayes *et al*, 2006; Di Fiore & Pines, 2010), the rate of YFP-Mcl-1 degradation was relatively slow (compare Fig 1D with Fig EV1E), as would be required for an apoptotic timer. Consistent with our previous results, YFP-Mcl-1 degradation in cells arrested in mitosis was inhibited by loss of APC/C activity (concomitant knockdown of the catalytic subunits APC2 and APC11) but was unaffected by depletion of Fbw7 (Fig 1D), even though the efficiency of knockdown was sufficient to stabilise protein levels of cyclin E and c-Myc, both well-documented Fbw7 substrates (Davis *et al*, 2014; Fig EV1F). Together, these results confirm that Mcl-1 degradation during mitotic arrest is not regulated by Fbw7 but is instead directed by the APC/C.

## A putative D box is a determinant of Mcl-1 degradation

We have identified a putative D box motif in Mcl-1, as well as a C-terminal IR tail, which is also found in the APC/C adaptor/activator proteins Cdc20 and Cdh1 (Harley *et al*, 2010; Fig 2A). Immediately C-terminal to the putative D box, we also noted a region containing charged residues including putative acceptor lysine residues that form a loosely defined sequence also found in other APC/C substrates. This region has been proposed to be a docking site for UbcH10, the initiating E2 for the APC/C (Williamson *et al*, 2011; Chang *et al*, 2015). Mutation of two key residues in the D box strongly inhibited YFP-Mcl-1 degradation (Fig 2B), consistent with our previous Western blotting analysis (Harley *et al*, 2010). Mutation of charged residues including two lysine residues C-terminal to the D box also stabilised YFP-Mcl-1, although to a lesser extent than the D box mutation (Fig 2C). The spacing between the D box and the lysines contained in this charged region suggests that they may

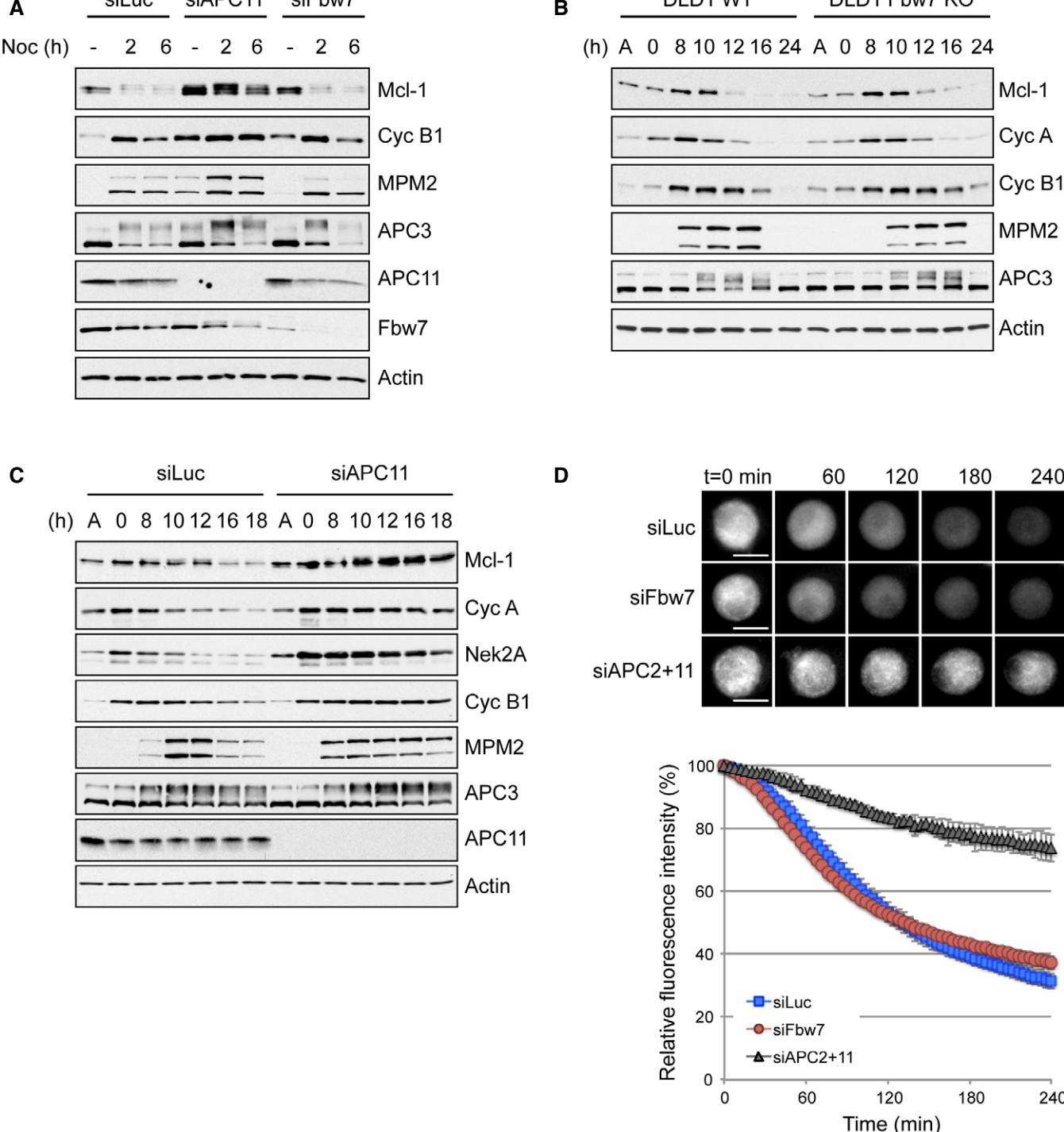

**Figure 1. Mcl-1 destruction during mitotic arrest is mediated by the APC/C, not SCF^Fbw7.**

A   HeLa cells transfected with control (siLuc), APC11 or Fbw7 siRNA for 48 h were synchronised in mitotic arrest with 830 nM nocodazole for 2 and 6 h and samples analysed with antibodies as indicated. An untreated sample (−) was used as a control.

B   DLD1 FBW7^+/+ (WT) and FBW7^−/− (KO) cells were released from a double thymidine block prior to addition of paclitaxel (500 nM). Samples were collected at times indicated after thymidine washout.

C   DLD1 cells WT for FBW7 transfected with control (siLuc) or APC11 siRNA were treated as in (B).

D   HeLa-YFP-Mcl-1 cells were transfected with siRNA as indicated and treated with nocodazole 48 h later. YFP-Mcl-1 fluorescence was monitored from the time of mitotic entry. Error bars represent ±SD, n = 3. Scale bar, 10 μm.

Source data are available online for this figure.

be targeted by UbcH10-APC/C for ubiquitination (Chang *et al*, 2015). Thus, the partial stabilisation exhibited by this mutant may be due either to loss of an UbcH10 docking site or loss of the optimal ubiquitin acceptor site(s). Nevertheless, these results attest further to the identity of the APC/C as an E3 ubiquitin ligase that targets Mcl-1 during mitotic arrest.

Deletion of the C-terminal IR motif (ΔIR) had no impact on the rate of YFP-Mcl-1 loss during mitotic arrest (Fig 2D), in contrast to Nek2A where a C-terminal methionine-arginine (MR) motif contributes to maximal Nek2A degradation during mitosis (Hayes *et al*, 2006). However, expression of the Mcl-1 D box mutant stabilised both cyclin A and cyclin B1 during mitotic arrest, as previously noted (Harley *et al*, 2010), and this was also the case for Nek2A degradation (Fig 2E), suggesting a dominant inhibitory effect on the APC/C. Remarkably, the ability of the Mcl-1 D box mutant protein to stabilise other substrates of the APC/C was lost when the IR motif was also deleted (DboxΔIR). Thus, although it is not necessary for Mcl-1 degradation, the IR motif appears to regulate the manner in which Mcl-1 engages with the APC/C to the exclusion of other substrates when the interaction is stabilised.

### The IR tail of Mcl-1 influences APC/C subunit requirements

The APC/C is a multisubunit enzyme that is activated by a co-factor, generally Cdc20 during mitosis and Cdh1 following mitotic exit. APC2 and APC11 form the catalytic core of APC/C while APC10 together with Cdc20 forms a receptor that recognises and binds to the D box of the substrate. Cdc20 and Cdh1, like Mcl-1, possess a C-terminal IR motif through which they bind to the scaffolding subunit, APC3. This is required for the substrate recruitment function of Cdc20/Cdh1. Cdc20 also binds to another scaffold subunit, APC8, to elevate the activity of APC/C (Chang *et al*, 2014, 2015). To examine the role of the IR motif in Mcl-1, we investigated the requirement for individual components of the APC/C in the degradation of wild-type (WT) and IR-deleted (ΔIR) YFP-Mcl-1 during a nocodazole arrest. Knockdown of either APC3 or APC11 stabilised YFP-Mcl-1 (WT Mcl-1; Figs 3A and EV2A). Similarly, YFP-Mcl-1ΔIR was stabilised by loss of APC11; however, unlike the wild-type protein, its degradation was hardly affected by APC3 knockdown (Figs 3B and EV2A). This suggests that the IR motif of Mcl-1 interacts with APC3 in a manner analogous to the IR tail of Cdc20 (Chang *et al*, 2014) and as suggested for the MR of Nek2A (Hayes *et al*, 2006), thus aiding recruitment of the protein to the APC/C. Interestingly, depletion of APC8 had no effect on the degradation of either protein (Fig 3A and B), which is consistent with the lack of stimulation by Cdc20 under these conditions. In contrast, and as expected, both cyclin A and cyclin B1 were stabilised by knockdown of each of the three APC/C subunits tested (Fig EV2B–D).

Interestingly, whereas degradation of WT YFP-Mcl-1 was insensitive, like cyclin A (Izawa & Pines, 2011), to loss of APC10 (Fig 3C), the degradation of YFP-Mcl-1ΔIR was retarded to an extent similar to that observed for cyclin B1 (Figs 3D, and EV2E and F). This suggests that APC10 is more important for Mcl-1ΔIR recruitment because loss of the IR-APC3 module invokes a greater reliance on the D box-APC10 interaction. Together, the differences in APC/C subunit requirement between WT Mcl-1 and Mcl-1ΔIR destruction indicate that the IR tail alters the nature of the interaction between Mcl-1 and the APC/C.

### The IR tail makes Mcl-1 degradation insensitive to the mitotic checkpoint

Although deletion of the IR motif had no discernible effect on YFP-Mcl-1 degradation during a strong mitotic arrest (Fig 2D), it did affect the rate of degradation upon exit from an unperturbed mitosis (Fig 4A). Fluorescence intensity declined sharply in YFP-Mcl-1ΔIR-expressing cells as they progressed through mitosis and continued as the cells underwent cytokinesis, whereas YFP-Mcl-1 declined more slowly and then became stabilised. IR-deleted Mcl-1 was, therefore, degraded more rapidly and completely than wild-type Mcl-1 protein when the checkpoint was switched off.

To investigate this further, we analysed the effect on Mcl-1 stability of inhibiting the checkpoint in cells arrested in mitosis by nocodazole. Under these conditions, the degradation of YFP-Mcl-1 was unaffected when the checkpoint was weakened by the Mps1 kinase inhibitor, reversine (Santaguida *et al*, 2010). Even in cells treated with a high dose of reversine, which caused them to exit mitosis rapidly, Mcl-1 degradation proceeded at a steady rate (Fig 4B). In stark contrast, the rate of YFP-Mcl-1ΔIR degradation increased dramatically as the checkpoint was weakened by increasing concentrations of reversine (Fig 4C). Thus, the APC/C-dependent degradation of Mcl-1 during mitosis is held in check by the IR tail. Without the interaction of the IR tail with the APC/C, Mcl-1 undergoes very rapid degradation when the checkpoint is switched off.

### The IR tail converts Mcl-1 from a Cdc20-dependent substrate to one that is degraded relatively slowly and independently of checkpoint control

Our finding that Mcl-1 degradation is unaffected by the spindle assembly checkpoint is unusual for an APC/C substrate, since the rates of loss of even those substrates such as cyclin A that are rapidly degraded by a Cdc20-dependent mechanism during mitotic arrest are still influenced by the checkpoint (Collin *et al*, 2013). We found that when Cdc20 was efficiently depleted by siRNA-mediated knockdown (Fig EV3A and B), YFP-Mcl-1 degradation and ubiquitination were not inhibited (WT Mcl-1; Figs 5A and EV3C). Rather there was a slight acceleration of YFP-Mcl-1 degradation when Cdc20 was removed. These results are in contrast to our previous biochemical experiments in which Cdc20 was depleted, albeit inefficiently (Harley *et al*, 2010), but in agreement with the lack of stabilisation of Mcl-1 following Cdc20 knockdown observed by others (Díaz-Martínez *et al*, 2014; Sloss *et al*, 2016). Indeed, we did not observe a requirement for an APC/C co-factor for Mcl-1 degradation since even efficient co-depletion of both Cdc20 and Cdh1 failed to inhibit the loss of YFP-Mcl-1 during mitotic arrest (Fig EV3D). In contrast, concurrent knockdown of APC2 and APC11 both prevented YFP-Mcl-1 ubiquitination (Fig EV3C) and stabilised the protein (Fig 1D). Similarly, under stringent checkpoint conditions, YFP-Mcl-1ΔIR degradation was also unaffected by knockdown of Cdc20 (Figs 5B and EV3E), despite the reduction in Cdc20 being sufficient to stabilise cyclin A (Fig EV3F).

Remarkably, however, the rapid degradation of YFP-Mcl-1ΔIR that occurred when the checkpoint was weakened was dependent upon Cdc20 such that prior Cdc20 knockdown slowed YFP-Mcl-1ΔIR degradation to a rate comparable to that of YFP-Mcl-1

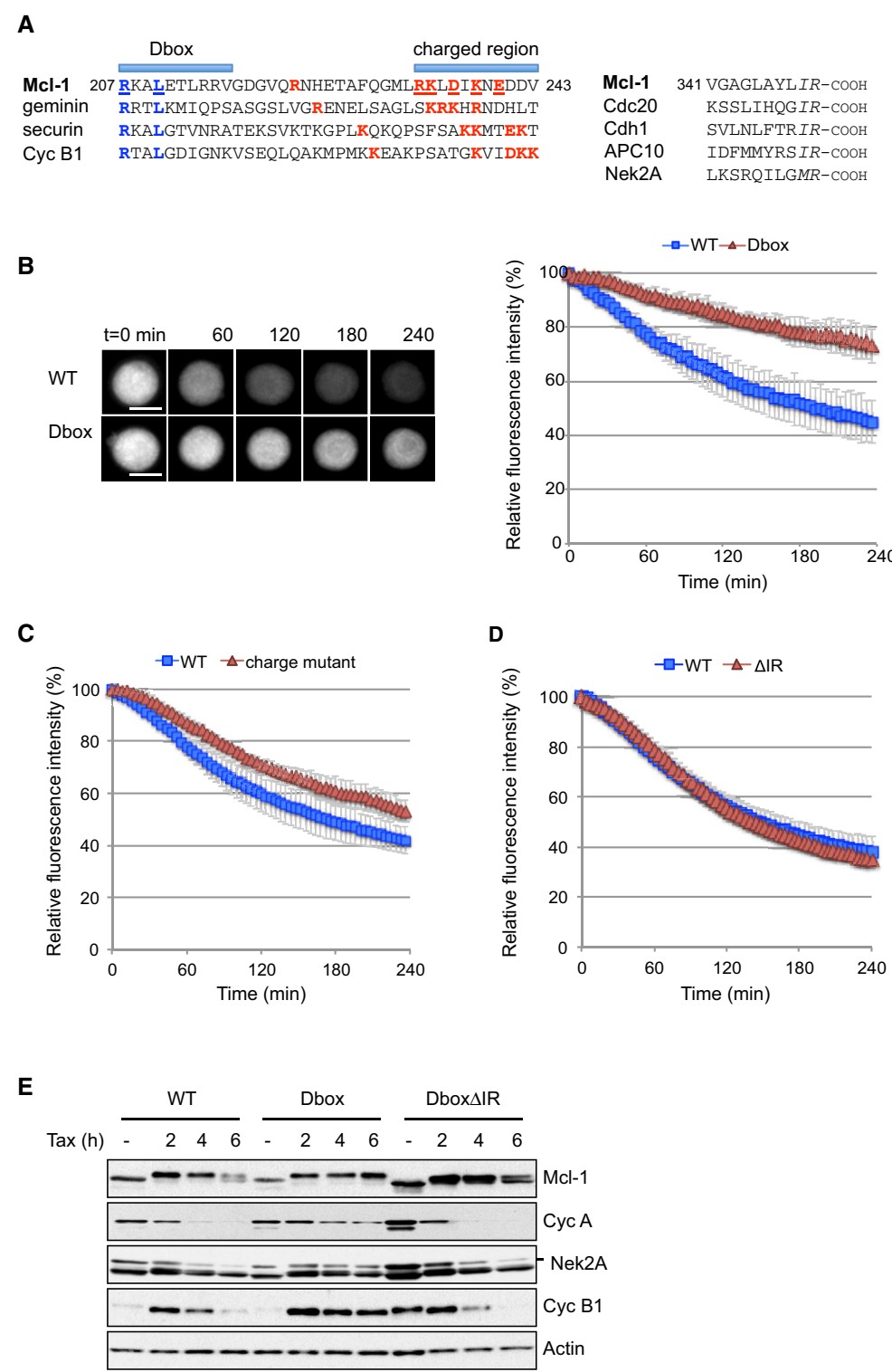

**Figure 2. The C-terminal IR motif is dispensable for Mcl-1 degradation.**

A       Sequence alignment of D box and IR motifs in Mcl-1 and other APC/C substrates. Key residues are indicated in the D box (blue) and C-terminal to the D box (red); those mutated in this study are underlined. The IR motif at the C-terminus is highlighted (italics).

B–D     Degradation of WT YFP-Mcl-1 protein compared with a D box mutant (B), charged residue mutant (C) and IR-deleted (ΔIR) Mcl-1 proteins (D) during mitotic arrest by time-lapse imaging. Error bars represent ±SD, *n* = 3. Scale bar, 10 μm.

E       U2OS cells expressing Flag-tagged WT, D box mutant or D box mutant ΔIR Mcl-1 proteins were synchronised in mitotic arrest with paclitaxel (100 nM). Samples were collected at times indicated and blotted with antibodies as shown.

Source data are available online for this figure.

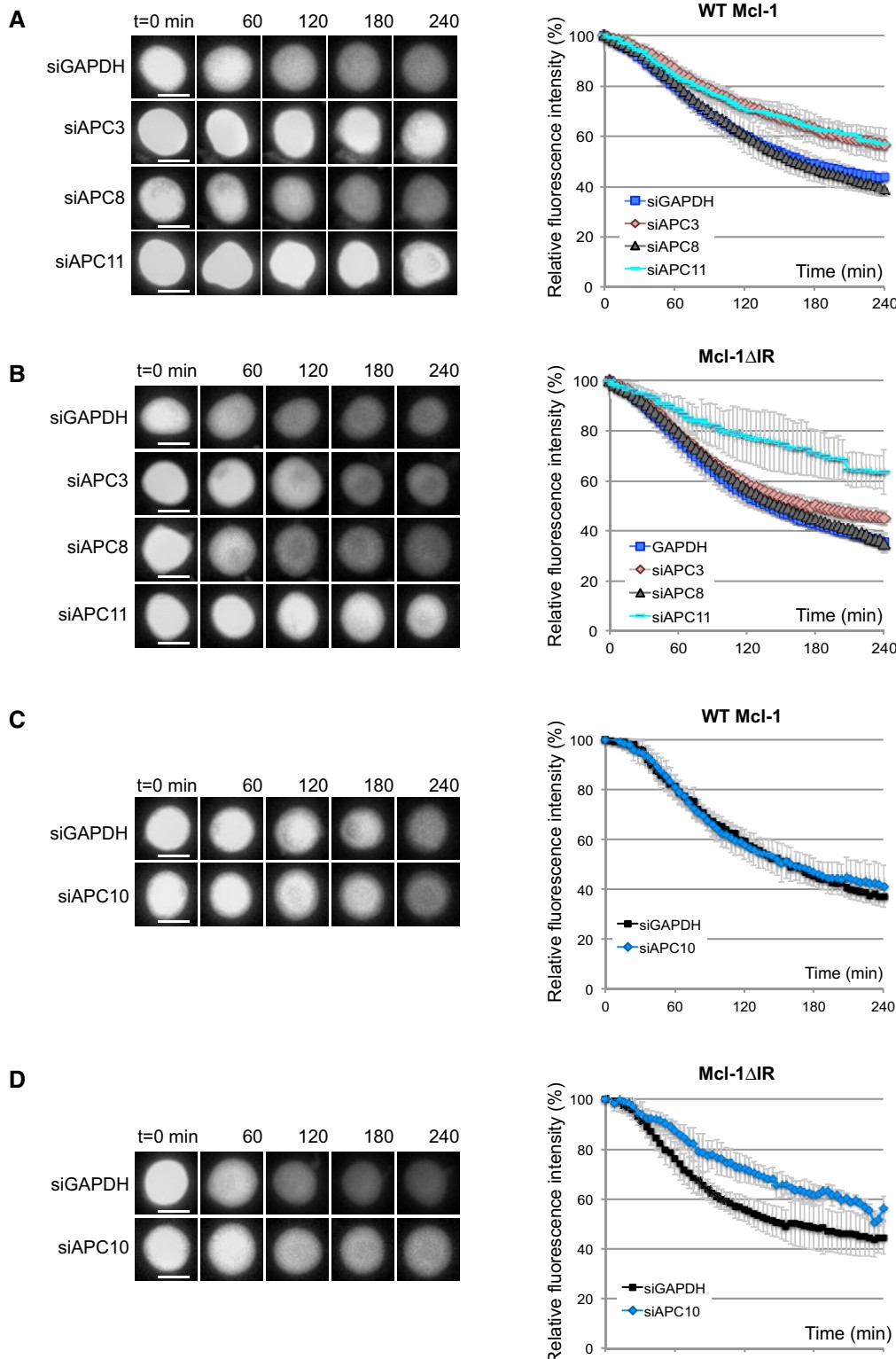

**Figure 3.　The IR tail alters APC/C subunit requirements for Mcl-1 degradation.**

A, B　APC3, APC8 or APC11 were knocked down by siRNA prior to treatment with nocodazole to induce mitotic arrest. The effect on the degradation of YFP-Mcl-1 WT (A) and ΔIR (B) protein was analysed by time-lapse microscopy. Representative images (left) and quantification (right) are shown. Error bars represent ±SD, n = 3. Scale bar, 10 μm.

C, D　The effect of knocking down APC10 on YFP-Mcl-1WT (C) and ΔIR (D) proteins during mitotic arrest is shown as for (A, B). Error bars represent ±SD, n = 3. Scale bar, 10 μm.

　　　　　　　　　　　

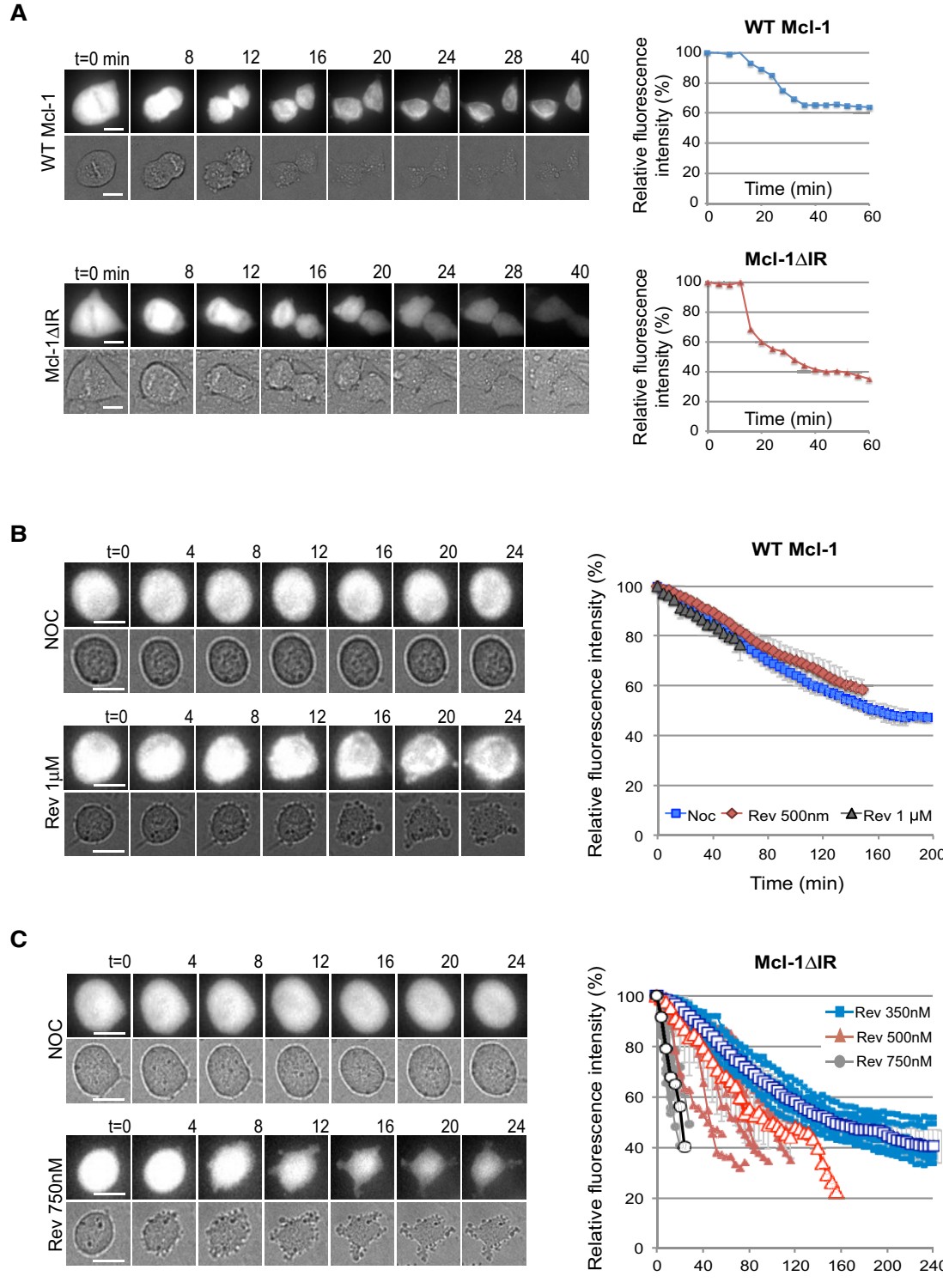

**Figure 4. The IR tail renders Mcl-1 degradation insensitive to the mitotic checkpoint.**

A    Representative images (left panels) and quantification (right panels) of the degradation of YFP-Mcl-1 WT and ΔIR proteins in unperturbed mitosis. Fluorescence (upper panels) and transmitted light (lower panels) images are shown. Scale bar, 10 μm.

B, C    Representative images (left panels) and quantification (right panels) of the degradation of YFP-Mcl-1 WT (B) and ΔIR (C) in cells entering mitosis in the presence of nocodazole and indicated concentration of reversine. In (B), mean values at each time point obtained from three experiments are shown. Error bars represent ±SD. In (C), quantification of 10 individual cells from one representative experiment is shown (right). The traces are overlaid with mean values ±SD (open characters, *n* = 3). Scale bar, 10 μm.

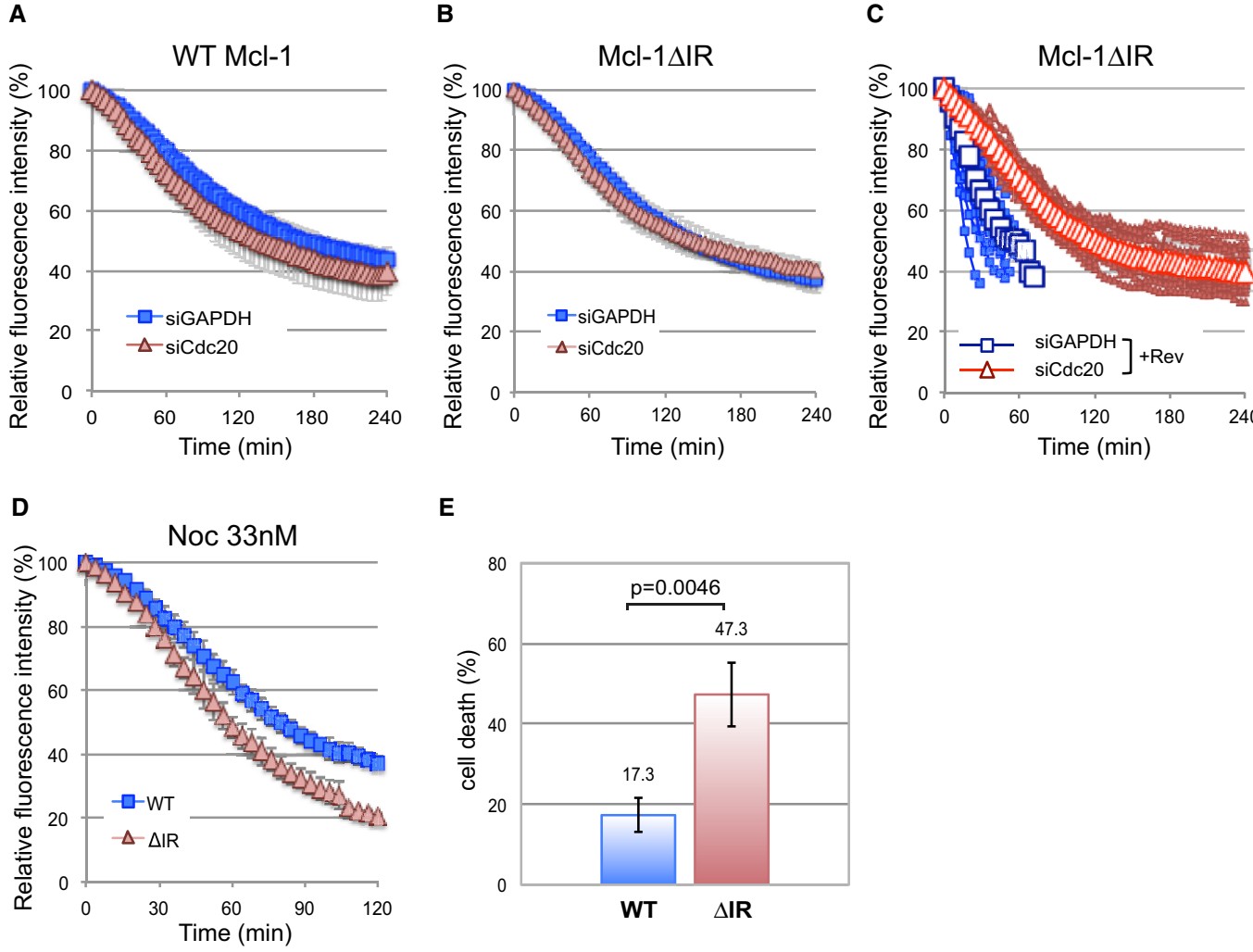

**Figure 5. The IR tail restrains Mcl-1 degradation to prevent unwarranted cell death.**

A, B    The degradation of YFP-Mcl-1 WT (A) and ΔIR (B) in nocodazole-arrested cells depleted of Cdc20 (siCdc20) or treated with a control siRNA (siGAPDH). Error bars represent ±SD, n = 3.

C    Degradation of YFP-Mcl-1ΔIR in cells depleted of Cdc20, entering mitosis in the presence of nocodazole and reversine (500 nM). Quantification of 10 individual cells from one experiment is shown. The traces are overlaid with mean values (open characters, n = 3). Error bars represent ±SD.

D    Comparison of the degradation of YFP-Mcl-1 WT and ΔIR in cells arrested under weak (33 nM nocodazole) checkpoint conditions. Error bars represent ±SD, n = 3.

E    Cell death in YFP-Mcl-1 WT and ΔIR-expressing cells depleted of endogenous Mcl-1 and arrested in mitosis under weak checkpoint conditions. For each experiment, the fate of 50 cells per condition was assessed and data were analysed using an unpaired, two-tailed Student's t-test.

Error bars represent ±SD, n = 3.

(Fig 5C). The rapid loss of YFP-Mcl-1ΔIR when the checkpoint was compromised was also dependent on APC3 (Fig EV3G). The effect of knocking down Cdc20 was not due simply to its ability to block mitotic exit, since YFP-Mcl-1ΔIR degradation continued even after mitotic exit induced by the CDK1 inhibitor RO-3306 (Vassilev *et al*, 2006) in control knockdown cells, whereas degradation was inhibited by Cdc20 depletion using siRNA (Fig EV3H). These results indicate a conventional mode of APC/C-mediated and Cdc20-dependent degradation of Mcl-1ΔIR when the checkpoint is switched off in which Cdc20 binds to APC3 and recruits Mcl-1ΔIR to the APC/C for ubiquitination.

One prediction from these results is that, under weak checkpoint conditions, cells expressing Mcl-1ΔIR would undergo apoptosis more readily because of the more rapid loss of anti-apoptotic activity. We specifically knocked down endogenous Mcl-1 using an siRNA duplex that targets the 3′ untranslated region (UTR), replaced it with either WT Mcl-1 or Mcl-1ΔIR expressed as fusions with YFP (Fig EV3I) and then arrested cells with a low concentration of nocodazole (33 nM). This resulted in rapid degradation of YFP-Mcl-1ΔIR (Fig 5D), whereas the rate of YFP-Mcl-1 (WT Mcl-1) destruction remained as slow as with a high nocodazole concentration (830 nM; compare Fig 5D with Fig 1D). Under such weak checkpoint conditions, < 20% WT Mcl-1 cells died. In marked contrast, almost half (47%) Mcl-1ΔIR-expressing cells underwent apoptosis (Fig 5E). This demonstrates that when the checkpoint is weak and Mcl-1 is degraded rapidly because of the

absence of the IR motif, extensive cell death is induced, whereas more stable wild-type Mcl-1 is still largely able to protect cells.

### The nature of the mitotic arrest determines cell fate

To explore further the relationship between Cdc20 and Mcl-1 degradation, we utilised proTAME, a cell-permeable molecule that is processed intracellularly to TAME (tosyl-L-arginine methyl ester), which mimics an IR dipeptide and binds to APC3, precluding the interaction of Cdc20 with the APC/C through its IR tail. This binding prevents the recruitment of Cdc20-dependent substrates to the APC/C and inhibits APC/C function, thereby arresting cells in mitosis (Zeng *et al*, 2010). We found that in cells held in mitosis after treatment with proTAME, YFP-Mcl-1 degradation was similar to that observed in cells arrested in mitosis by nocodazole (Fig 6A). This result is in agreement with Sloss *et al* (2016), who found that apcin, another reagent that inhibits the stimulation of the APC/C by Cdc20 (Zeng *et al*, 2010; Sackton *et al*, 2014), did not block Mcl-1 destruction. Nevertheless, YFP-Mcl-1 was stabilised in cells arrested by knockdown of APC2 and APC11, consistent with our previous observations in the presence of nocodazole (compare Fig 6A with Fig 1D). Together, these results are consistent with APC/C-mediated Mcl-1 destruction during mitotic arrest being either independent of Cdc20 or unusually sensitive to very low levels of the APC/C co-activator. However, the inability of proTAME to inhibit YFP-Mcl-1 loss even when combined with Cdc20 depletion (Fig EV4A) favours the conclusion that APC/C-dependent Mcl-1 degradation during mitotic arrest does not require the stimulation of the APC/C by Cdc20.

In contrast to YFP-Mcl-1, degradation of cyclin B1-Venus was negligible during an arrest induced with either proTAME or depletion of APC2 and 11 (Fig 6B). These results demonstrate that two distinct modes of inducing mitotic arrest have differential effects on the relative rates of cyclin B and Mcl-1, yielding populations of cells with different relative levels of these two proteins. Given that the outcome of a mitotic arrest is likely to be co-ordinately regulated by apoptotic and mitotic thresholds (Gascoigne & Taylor, 2008; Clarke & Allan, 2009), this raises the intriguing possibility that the nature of a mitotic arrest may influence cell fate. To investigate this, we compared cell fate profiles of RPE-1 cells arrested either with proTAME or by concomitant knockdown of APC2 and 11 (Fig EV4B). Under these conditions, the duration of mitotic delay was similar, negating the influence of time in mitosis on cell fate (Fig 6C). Analysis was restricted to cells showing a sustained arrest (≥ 6 h). Consistent with previous results using HeLa cells (Zeng *et al*, 2010; Lara-Gonzalez & Taylor, 2012), RPE-1 cells arrested with proTAME predominantly underwent mitotic cell death (80%) after arresting for an average of 24 h, while the remaining 20% slipped out of mitosis after an average arrest of 32 h (Fig 6C, top left panel). In comparison, knockdown of APC2 and APC11 substantially altered the cell fate profile with nearly 30% of cells exhibiting a sustained mitotic arrest (average 41 h) while some were still arrested at the end of imaging (Fig 6C, top right panel). Indeed, even those cells that did undergo mitotic cell death did so after a more prolonged arrest (average 29 h). Notably, this prolonged cell survival was dependent upon Mcl-1 since knockdown of Mcl-1 converted the cell fate profile exclusively to mitotic cell death with an average arrest time prior to death of

19 h (Fig 6C, bottom right panel). Similarly, prior Mcl-1 depletion further promoted mitotic cell death in response to proTAME (Fig 6C, bottom left panel). These results suggest that the degradation of Mcl-1 during a proTAME-induced arrest promotes mitotic cell death, whereas the relative stability of Mcl-1 following depletion of APC2 and APC11 supports cell survival. Thus, the nature of the mitotic arrest determines cell fate, at least in part, through its effect on Mcl-1 protein levels.

## Discussion

We have confirmed that, during mitotic arrest, the degradation of Mcl-1 is directed primarily by the APC/C, an ubiquitin E3 ligase that regulates many of the proteins involved in mitotic progression. Utilising the same experimental systems as reported previously (Wertz *et al*, 2011), we have demonstrated that SCF^Fbw7 does not influence Mcl-1 protein levels in cells during mitotic arrest. However, loss of Fbw7 does affect the ability of cells to arrest in mitosis (Finkin *et al*, 2008; Wertz *et al*, 2011; this study) and thus may appear to stabilise Mcl-1 in a biochemical analysis unless cells slipping out of mitosis are excluded. SCF^Fbw7 might nevertheless play a role in Mcl-1 degradation after slippage into interphase and thereby influence subsequent cell fate (Inuzuka *et al*, 2011; Wertz *et al*, 2011). Other E3 ubiquitin ligases, namely Mule/ARF-BP1/HUWE (Zhong *et al*, 2005; Shi *et al*, 2011) and SCF^ßTRCP (Ding *et al*, 2007), have been reported to mediate the degradation of Mcl-1 during interphase. Degradation of a mutant lacking lysine ubiquitin acceptors has suggested that E3 ligase-independent degradation can also occur (Sloss *et al*, 2016). Our results, however, demonstrate that APC/C-mediated degradation is the major mechanism controlling Mcl-1 loss during mitosis.

The degradation of Mcl-1 over a period of several hours fulfils the requirements of an apoptotic timer during mitotic arrest. The APC/C dependency of Mcl-1 destruction couples the initiation of its loss to entry into mitosis. Unusually for an APC/C substrate, however, the mitotic destruction of Mcl-1 is not influenced by either checkpoint status or known co-factors that enhance APC/C activity, resulting in a steady loss of Mcl-1 protein that ceases only when the cell exits mitosis. We propose that the relatively slow rate of Mcl-1 destruction is determined by an atypical mechanism of recognition by the APC/C such that the complex is insensitive to stimulation by Cdc20 and the degradation of Mcl-1 is not induced like that of cyclin B1 when the checkpoint is relieved. Otherwise, rapid degradation of Mcl-1 when the checkpoint is weakened or switched off would result in catastrophic loss of Mcl-1 and apoptosis (Fig 7A, left). Under weak checkpoint conditions, as occur during the normal progression of prometaphase/metaphase when the checkpoint is maintained for a short period by perhaps only one or two unattached kinetochores (Rieder *et al*, 1995; Collin *et al*, 2013), the limited destruction of wild-type Mcl-1 that occurs prior to mitotic exit would not compromise cell viability (Fig 7A, middle). Under conditions of a prolonged, strong checkpoint arrest when mitosis is dramatically disrupted by a failure to form the mitotic spindle, for instance due to the action of a cancer chemotherapeutic microtubule poison, the steady rate of Mcl-1 destruction would still result in apoptosis once a threshold level was reached because its degradation is insensitive to the

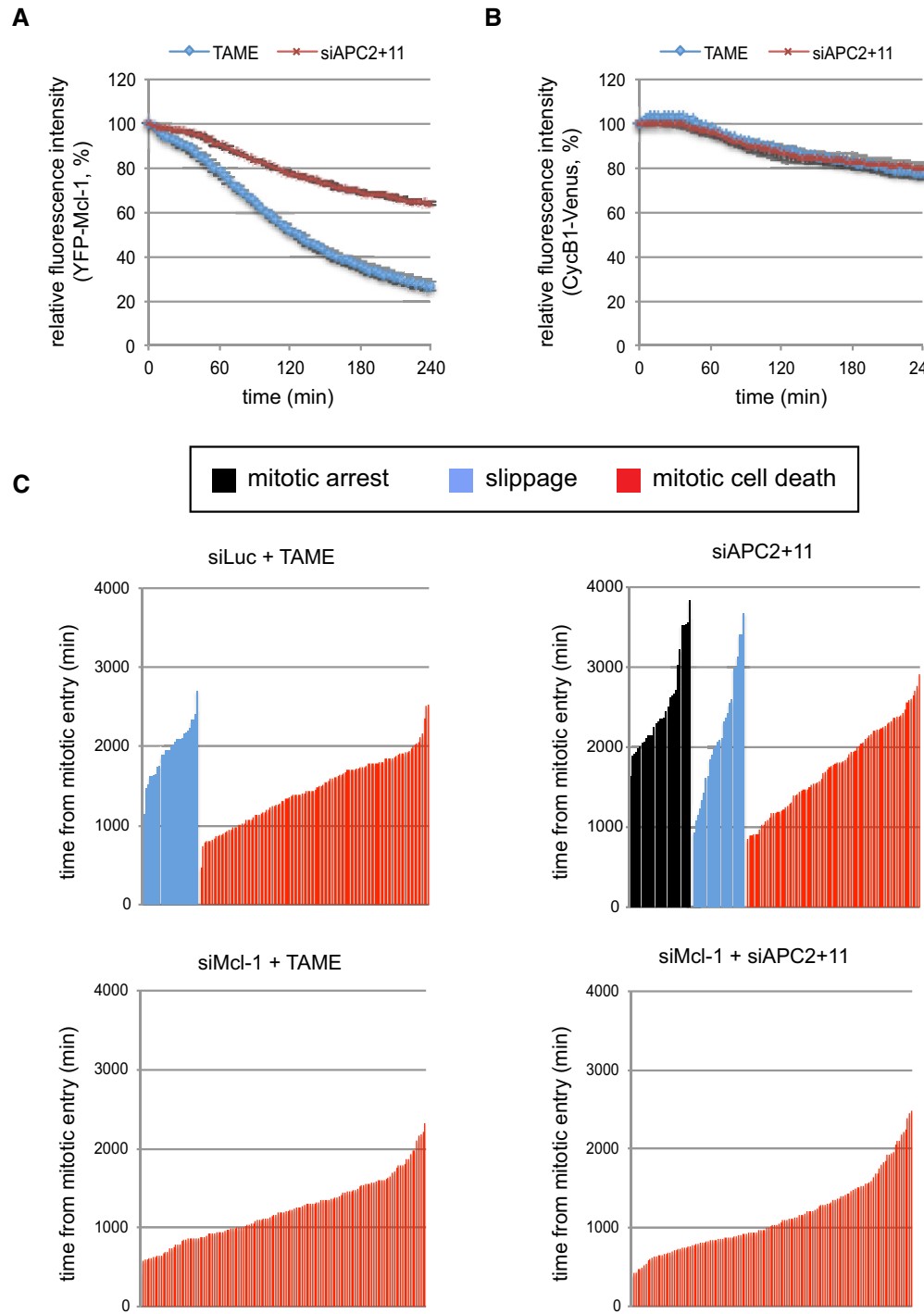

**Figure 6. The mode of mitotic arrest alters Mcl-1 destruction and determines cell fate.**

A, B    Comparison of the degradation of YFP-Mcl-1 WT (A) and CycB1-Venus (B) in cells arrested in mitosis either by treatment with proTAME (10 μM) or by co-depletion of APC2 and APC11. The trace shows the average of three experiments. Error bars represent ±SD, $n = 3$.

C    Cell fate profiles are shown for RPE cells arrested in mitosis either by treatment with proTAME (10 μM) or by co-depletion of APC2 and APC11 (upper panels). The effect on cell fate of depleting Mcl-1 concurrently is shown (lower panels). The combined data from three independent experiments are shown ($n \geq 140$ cells).

checkpoint even though the rate of degradation of cyclin B1 and other mitotic APC/C substrates that hold a cell in mitosis is reduced (Fig 7A, right). Thus, the atypical mechanism of APC/ C-dependent Mcl-1 destruction ensures that the initiation of apoptosis during mitotic arrest is dependent on the period of mitosis and not the degree of checkpoint activation.

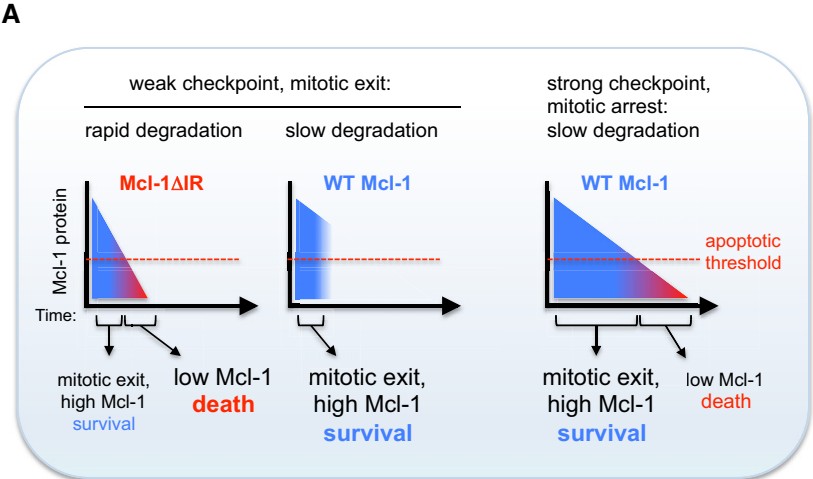

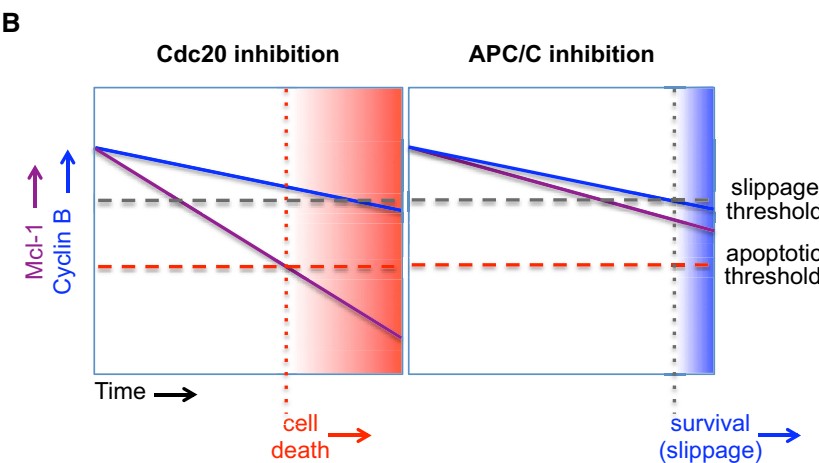

**Figure 7. Model of the effect of checkpoint strength and nature of mitotic arrest on cell fate.**

A   Model of the role of the Mcl-1 IR motif during mitotic arrest under conditions of different checkpoint strength.

B   The nature of mitotic arrest determines cell fate through differential effects on Mcl-1 protein degradation.

Taken together, the checkpoint insensitivity of Mcl-1 degradation, the lack of dependency on APC8 and the lack of inhibition by an efficient knockdown of Cdc20 and/or treatment with TAME all indicate that Mcl-1 is usually ubiquitinated during mitotic arrest by the APC/C unstimulated by Cdc20. The contrasting rapid, Cdc20-dependent degradation of Mcl-1ΔIR when the checkpoint is released argues against the alternative explanation that Mcl-1 recognition by the APC/C requires only a trace of Cdc20. Rather, it appears that stimulation by Cdc20 of APC/C activity towards Mcl-1 is usually prevented by the competing interaction of the IR tail of Mcl-1 with the APC/C, but in the absence of this interaction, Mcl-1 behaves as a conventional APC/C^Cdc20 substrate. Despite its usual lack of dependency on Cdc20, Mcl-1 degradation is, however, still dependent on its D box motif, which suggests that the D box is required for correct alignment of Mcl-1 with the APC/C even in the absence of the co-factor or it plays a role in the release of the substrate from the enzyme. Indeed, the dominant inhibitory effect of the D box mutant of Mcl-1 on the degradation of other APC/C substrates

suggests that this mutant forms a more stable interaction with the APC/C than the wild-type protein.

The unusual mode of functional interaction of Mcl-1 with the APC/C may provide a mechanism for regulation of Mcl-1 degradation during mitosis. For instance, phosphorylation of Mcl-1 by Cdk1-cyclin B, which promotes Mcl-1 degradation (Harley *et al*, 2010), might affect the interaction of the IR tail with the APC/C, thereby promoting Mcl-1 degradation through a Cdc20-stimulated mechanism. This could occur when CDK1-cyclin B activity is high but the spindle assembly checkpoint is weak. Such conditions might explain why we previously observed a dependency of Mcl-1 loss on Cdc20 in experiments in which Cdc20 depletion was partial and the cells were prone to slippage from mitosis (Harley *et al*, 2010). The binding of a partner protein such as Noxa could also affect the interaction of Mcl-1 with the APC/C and control Mcl-1 stability (Nakajima *et al*, 2014; Haschka *et al*, 2015). Conversely, although the interaction of the APC/C with Mcl-1 is likely to be normally sufficiently transient to not interfere

with the recognition of other substrates such as cyclin B, there is the potential for this interaction to significantly affect APC/C function when its activity is limited or when Mcl-1 is highly expressed, for instance in certain cancer cells. Indeed, depletion or over-expression of Mcl-1 has been reported to affect cyclin B1 destruction and the timing of exit from mitosis (Sloss *et al*, 2016). Thus, Mcl-1 might play an indirect role in influencing progression through mitosis as well as the onset of mitotic cell death.

We have demonstrated that the atypical mode of Mcl-1 degradation during mitosis can be exploited to influence cell fate. Utilising proTAME to arrest cells in mitosis while still permitting Mcl-1 destruction promotes cell death. In contrast, a strategy of targeting the APC/C directly stabilises Mcl-1 during the arrest and supports cell survival (Fig 7B). Previously, it has been shown that the duration of the mitotic arrest correlates with the likelihood that a cell will undergo apoptosis (Bekier *et al*, 2009; Huang *et al*, 2009; Colin *et al*, 2015). Furthermore, targeting Cdc20 was suggested as more efficient than spindle poisons at inducing apoptosis because it blocked mitotic exit in slippage-prone cells (Huang *et al*, 2009). Our data now provide an explanation for these effects. Cdc20 loss does not simply prevent slippage and increase the time spent in mitotic arrest; crucially, cells arrested in this way continue to degrade Mcl-1. As a result, the apoptosis threshold is overcome while cells are still arrested in mitosis (Fig 7B). Thus, cell fate is determined by the nature as well as the duration of mitotic arrest. Inhibiting Cdc20, therefore, offers a therapeutic opportunity to specifically target Mcl-1 for destruction and engage the apoptotic machinery during a protracted mitosis. Conversely, Mcl-1 would be stabilised and apoptosis inhibited by a strategy targeting the APC/C directly, which might have less desirable consequences for cancer chemotherapy.

If TAME inhibits the interaction of the IR tail of Mcl-1 with the APC/C, concurrent interference with the stimulation of the APC/C by Cdc20 is presumably the reason that TAME does not accelerate Mcl-1 destruction during mitosis. Therefore, an improved strategy might be to design a selective inhibitor of the interaction between the C-terminal tail of Mcl-1 and the APC/C, which would mimic the IR deletion and accelerate the Cdc20-dependent degradation of Mcl-1 to actively promote death in cells arrested in mitosis by combination with a microtubule poison or other anti-mitotic drug.

# Materials and Methods

### Antibodies and reagents

The following antibodies were used for Western blotting: APC11 (1/500, ab57158) and MYC (1/5,000, ab32072), Abcam; APC3 (1/1,000, 610454), cyclin A (1/1,000, 611269), Mcl-1 (1/500, 559027) and Nek2A (1/500, 610593), BD Transduction Laboratories; APC8 (1/1,000, A301-181A) and Fbw7 (A301-720A-1), Bethyl Laboratories; APC2 (1/1,000, 12301S) and cyclin E (1/1,000, 4129S), Cell Signaling Technologies; MPM2 (1/1,000, 05-368), Millipore; APC10 (1/1,000, sc-166790), Cdc20 (1/1,000, sc-13162) and cyclin B1 (1/2,000, sc-752), Santa Cruz Biotechnology; Actin (1/4,000, A2066) and Mad2 (1/1,000, PA5-21594), Sigma Aldrich; Cdh1 (1/500,

MS1116-P0), Thermofisher. Antibodies were used according to standard protocols except for Fbw7 where blocking and primary antibody incubation were carried out in ReliaBLOT (Bethyl Laboratories). Histone H3 phospho-Ser10 (05-806), Millipore, was used for flow cytometry as described previously (Harley *et al*, 2010). Fbw7 (A301-721A-1), Bethyl Laboratories, was used for immunoprecipitation. Doxycycline, thymidine, paclitaxel and reversine were purchased from Sigma Aldrich. Nocodazole (Calbiochem) was routinely used at 830 nM apart from assaying cyclin B1 degradation where it was used at 33 nM. ProTAME was initially provided by Dr Randall King (Dept. of Cell Biology, Harvard University) and subsequently purchased from Boston Biochem.

### Cell lines and treatments

Authenticated HeLa (OHIO) and U2OS (HTB96) were obtained from Cell Services, Cancer Research UK London Research Institute, RPE-1 cells were from ATCC, and DLD1 parental and FBW7$^{-/-}$ cells (Wertz *et al*, 2011) were purchased from Horizon Discovery Ltd. HeLa Flp-In cells (Tighe *et al*, 2008) stably expressing a TetR were from Prof Stephen Taylor (Manchester). All cells were cultured in DMEM, 10% FBS, 2 mM L-glutamine and 50 μg/ml penicillin/streptomycin. Cells were routinely tested for mycoplasma contamination. The generation of U2OS cells expressing Flag-Mcl-1 WT or Flag-Mcl-1 D box mutant was described previously (Harley *et al*, 2010). U2OS Flag-Mcl-1 D box mutant ΔIR cells expressing Mcl-1 R207A/L210A/C-terminal 2 amino acids (IR) deleted were generated in the same way. Flp-In cells stably expressing a doxycycline-inducible construct were generated by co-transfection of pOG44 (Invitrogen) and cDNAs cloned into pc5-LAP-YFP or pc5-LAP-Venus. Stable polyclonal cell lines were selected in media supplemented with Hygromycin (200 μg/ml). Doxycycline (1 μg/ml) was added to inducible cell lines for 8 h to induce protein expression prior to imaging. The protocol for synchronising cells in the period of mitotic arrest with nocodazole was described previously (Harley *et al*, 2010). For synchronisation of DLD1 cells, thymidine (2 mM) was added for 16 h followed by incubation in thymidine-free media for 8 h prior to incubation with thymidine for a further 16 h. Cells were then released into normal media for 6 h prior to addition of paclitaxel to arrest cells in mitosis.

### Vectors

For Mcl-1 constructs to generate inducible cell lines, AgeI (5′) and ApaI (3′) restriction sites were introduced by PCR using WT or D box or ΔIR mutant Flag-Mcl-1 vectors as templates. The resulting Mcl-1 cDNA was used to replace Mps1 in pc5-LAP-YFP-Mps1 (Nijenhuis *et al*, 2013) C-terminal to YFP. The YFP-Mcl-1 charge mutant R233A/K234A/D236A/K238A/E240A was generated by site-directed mutagenesis using pc5-LAP-YFP-WT Mcl-1 as a template. Nek2A was amplified by RT–PCR and cloned into pc5-LAP-YFP vector as for Mcl-1. To generate pc5-cyclin A-Venus, cyclin A-Venus was isolated from pc4-cyclin A-Venus using HindIII and ApaI and used to replace cyclin B1-mCherry in pc5. To generate pc5-cyclin B1-Venus, Venus cDNA was isolated from pc4-cyclin A-Venus using BamHI and ApaI. This was used to replace mCherry in pc5-cyclin B1-mCherry. Cyclin constructs were provided by Dr Adrian Saurin (Dundee). pMT.107-(His6Ub)8 was provided by Dr David Meek (Dundee).

**Transfections**

To generate stable inducible cell lines, vectors were co-transfected with pOG44 (encodes a FLP recombinase) into HeLa Flp-In cells stably expressing a TetR using FuGene HD according to the manufacturer's instructions (Promega). To knockdown protein expression, ON-TARGETplus siRNAs from Dharmacon were used unless otherwise stated: Fbw7-1 GGGCACCAGUCGUUAACAA (J-004264-07), Fbw7-3 GGAGUUGUGUGGCGGAUCA (J-004264-09) and Fbw7-4 CAACAACGACGCCGAAUUA (J-004264-10); APC2-2 GAUCGUAUCUACAAC AUGC (J-003200-12), APC2-3 GACAUCAUCACCCUCUAU (J-003200-11) and APC2-4 GAGAUGAUCCAGCGUCUGU (J-003200-10); APC3-1 GGAAAUAGCCGAGAGGUAA (J-003229-11) and APC3-2 CAAAAGAGCCUUAGUUUAA (J-003229-12); APC8-1 GAAAUUAAAUCCUCGGUAUUU and APC8-2 GCAGUUGCCUAUCACAAUAUU (Izawa & Pines, 2011; Eurofins-MWG); APC10-1 GAGCUCCAUUGGUAAAUUUUU and APC10-2 GAAAUUGGGUCACAAGCUGUU (Izawa & Pines, 2011; Eurofins-MWG); APC11-1 (UCUGCAGGAUGGCAUUUAA and APC11-2 AAGAUUAAGUGCUGGAACG (Mansfeld *et al*, 2011; Eurofins-MWG); Cdc20-1 CGGAAGACCUGCCGUUACA (J-003225-14), Cdc20-2 GGGCCGAACUCCUGGCAAA (J-003225-15), Cdc20-3 GAUCAAAGAGGGCAACUAC (J-003225-16) and Cdc20-4 CAGAACAGACUGAAAGUAC (J-003225-17); Cdh1 5′-AAU GAG AAG UCU CCC AGU CAG-3′ (Bashir *et al*, 2004); Mcl-1 UTR CGAAGGAAGUAUCGAAUUU (J-004501-17); GAPDH AUUCCAUGGCACCGUCAAG and Luciferase GL2 CGUAC GCGGAAUACUUCGA (Eurofins-MWG). siRNA duplexes for each target were pooled and transfections carried out using 10–20 nM final duplex concentration and LipofectAmine RNAiMAX (Invitrogen) according to the manufacturer's instructions for 48 h apart from for Cdc20 which was carried out for 24 h.

**Live-cell imaging**

To study fluorescent protein stability by time-lapse analysis, cells were plated into eight-well chamber slides (Ibidi). For knockdown experiments, cells were treated for 24 h with thymidine (2 mM) before being released into Leibovitz L-15 medium (Sigma), supplemented with 10% FBS, 2 mM L-glutamine, 100 μg/ml penicillin/streptomycin and 1 μg/ml doxycycline. For other experiments, plating media was replaced with L-15 and doxycycline as above. After 4 h, nocodazole was added and, a further 4 h later, cells were imaged in a heated chamber (37°C), using Delta Vision Core or Elite microscope system equipped with a 40×/1.30 NA U Plan FLN objective using softWoRx software. Images were acquired with a CoolSNAP HQ2 camera (Photometrics) and processed using softWoRx software and ImageJ Fiji. Stacks of five images at 6-μm intervals were taken at 4 min intervals for 16–18 h. Maximum intensity projection of the fluorescent channels was performed using softWoRx software. In all experiments assessing unperturbed mitosis or mitosis with a weak spindle assembly checkpoint, a brightfield reference image was also captured to visualise cell morphology. Images shown were selected to most closely represent the mean data. For analysis of cell fate, cells were cultured in 24-well plates (Thermo Scientific) in DMEM supplemented with 10% (v/v) foetal bovine serum and 1% (v/v) penicillin-streptomycin (Invitrogen). Cells were imaged every 10 min in a heated chamber (37°C, 5% $CO_2$) using an EC Plan-Neofluar 10×/0.3 objective on a Zeiss Axiovert 200M

microscope. Images were acquired using a C4742-80-12AG camera (Hamamatsu) and μmanager software and were processed using ImageJ Fiji. Cell death was defined by morphology and cessation of cell movement. For the determination of cell fate in RPE cells, analysis was carried out on cells that were arrested in mitosis for at least 6 h.

**Analysis of Mcl-1 ubiquitination**

YFP-Mcl-1 cells were transfected with siRNA against APC2 and APC11 (48 h) or Cdc20 (24 h) and with pMT.107-(His$_6$Ub)8 or control vector. Dox was added to induce expression of YFP-Mcl-1 for 18 h prior to addition of nocodazole (0.83 μM) and MG132 (10 μM) for 4 h. Mitotic cells were isolated and divided into two as follows: for input samples, cells were lysed in SDS loading buffer (50 mM Tris, pH 6.8; 2% SDS; 10% glycerol) and the protein concentration was determined; for pulldowns, cells were lysed in guanidine-HCl buffer (6 M guanidine-HCl; 10 mM Tris, pH 8.0; 0.1 M Na$_2$HPO$_4$; 5 mM imidazole) with the volume dictated by the concentration of the input samples. Lysates were sonicated and incubated with nickel-NTA agarose beads at room temperature for 4 h. Pulldowns were washed with urea wash buffer (8 M urea; 0.1 M Na$_2$HPO$_4$) sequentially containing A (0.1% Triton X-100; 10 mM Tris, pH 8.0), B (0.2% Triton X-100; 10 mM Tris, pH 6.5) and C (0.1% Triton X-100; 10 mM Tris, pH 6.5). Proteins were eluted in elution buffer (250 mM imidazole; 5% 2-mercaptoethanol; 50 mM Tris, pH 6.8; 2% SDS; 10% glycerol).

**Expanded View** for this article is available online.

### Acknowledgements

This study was funded by Cancer Research UK (C275/A13424 to PRC) and Worldwide Cancer Research (formerly the Association for International Cancer Research; 12-0105 to PRC). KIR was supported by a Biotechnology and Biological Sciences Research Council (BBSRC) EastBio Programme Studentship. We are grateful to Dr Adrian Saurin for reagents and discussion.

### Author contributions

LAA and AS carried out the experiments. KIR contributed data to Fig EV2 and DZ generated reagents. LAA and PRC wrote the manuscript. PRC supervised the study.

### Conflict of interest

The authors declare that they have no conflict of interest.

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
