## [Review Process File · The EMBO Journal]

Atypical APC/C-dependent degradation of Mcl-1 provides an apoptotic timer during mitotic arrest.

Lindsey A. Allan, Agnieszka Skowyra, Katie I. Rogers, Désirée Zeller and Paul R. Clarke

Review timeline:

Submission date:	27 th February 2017
Editorial Decision:	3 rd May 2017
Revision received:	21 st December 2017
Editorial Decision:	9 th March 2018
Revision received:	3 rd May 2018
Accepted:	16 th May 2018

Editor: Anne Nielsen.

Transaction Report:

1st Editorial Decision

3rd May 2017

Thank you for submitting your manuscript for consideration by the EMBO Journal and my apologies for the extended duration of the review process. Your study has now been seen by three referees whose comments are shown below.

As you will see from the reports, all three referees express interest in the findings reported in your manuscript and highlight the importance of settling the question of Mcl-1 degradation in mitosis. However, they also raise a number of major and minor concerns that need to be sorted out in order to conclusively support the notion that APC/C is directly responsible for Mcl-1 degradation. In our view, these are reasonable and constructive points that would strengthen the manuscript and which should be addressed before we can offer publication in The EMBO Journal.

Given the referees' overall positive recommendations, I would like to invite you to submit a revised version of the manuscript, addressing the comments of all three reviewers. I should add that it is EMBO Journal policy to allow only a single round of revision, and acceptance of your manuscript will therefore depend on the completeness of your responses in this revised version.

For the revised manuscript I would particularly ask you to focus your efforts on the following points:

-> Please elaborate on the discrepancies reported for the APC/C and Cdc20 contribution in your own earlier work, Wertz et al 2011, Diaz-Martinez et al, and the present study (ref #2 point 1), in order to place your findings adequately in the context of the published literature

-> Refs #2 (point 2) and ref #3 (point 2) both find that the model for APC/C-dependent ubiquitination of Mcl-1 requires additional experimental data and extensive rewriting/discussion to make the findings more accessible for a non-specialised audience. In our view, this is a valid point that would also go some way to address the concern from ref #1 that the current study remains 'a little undefined in mechanism'

-> Regarding in vitro reconstitution of APC/C-Mcl-1 (ref #2) and timelapse movies of Mcl-1 degradation in cells (ref #3) I realize that these may not be trivial experiments and I'd be happy to discuss what kind of data you would be able to include to address these points.

REFEREE REPORTS.

Referee #1:

In this study the authors have investigated the proteolysis of the Mcl1 protein in mitosis. Using FBW7 knockout cells, they find that Mcl1 is targeted by the APC/C and not by SCF-FBW7 as previously reported. This is an important finding as it settles a contentious issue. The authors go on to show that Mcl1 is an unusual APC/C substrate in that it requires little Cdc20 for its degradation and is steadily degraded through mitosis, even when the Spindle assembly checkpoint (SAC) is satisfied. The authors conclude that this behaviour allows Mcl1 degradation to act as a 'mitotic timer' for the apoptotic machinery. The authors show that this behaviour is conferred on Mcl1 by its Isoleucine-Arginine tail that binds to the APC/C. Mcl1 lacking its IR tail is degraded rapidly when the SAC is inactivated. Overall these findings are intriguing - if a little undefined in mechanism - and should be of interest to the readers of The EMBO Journal.

Before publication I think the authors should address the following points:

1) The siRNA experiments of APC8 and Cdc20 are useful to show the difference in sensitivity to the level of the proteins but I would caution the authors against pushing their conclusions too far. From the published structures of the APC/C it is difficult to conceive of an APC/C that lacks APC8; similarly it is difficult to envisage how a protein other than one with a C-box could activate the APC/C in the absence of Cdc20 or Cdh1. My interpretation of these depletion experiments is that the authors have not removed APC8 and Cdc20 completely and that Mcl1 is simply a better substrate because it can bind directly to the APC/C.

2) The authors state that TAME binds to Cdc20, but TAME mimics an IR dipeptide and thus binds to the IR-receptor on APC3 (doi.org/10.1016/j.ccr.2010.08.010). Thus TAME is likely to be in competition with Mcl1 to bind to the APC/C and if Mcl1 affinity for the APC/C is higher this could explain why TAME does not stabilise Mcl1, rather than though the effect of TAME on Cdc20 binding.

Minor point: In Fig 5c have the points been mis-labelled? It appears that Cdc20 knockdown causes faster Mcl1 degradation.

Referee #2:

Mitotic cell death following perturbation of mitosis is failsafe mechanism to prevent the emergence of chromosomal abnormalities. Activation of the spindle checkpoint followed by induction of mitotic cell death by antimetabolites is a major strategy in cancer treatment. A critical target of the checkpoint is CDC20, the activator of the anaphase-promoting complex/cyclosome (APC/C), which is a multi-subunit ubiquitin ligase complex that initiates mitotic exit. MCL-1 is an anti-apoptotic BCL-2 family protein that suppresses the activation of caspases. MCL-1 is gradually degraded by ubiquitin-mediated proteolysis during prolonged mitotic arrest. Apoptosis is initiated, when the levels of MCL-1 decline to below the critical threshold. Thus, MCL-1 degradation has been proposed to act a timing device for mitotic cell death. Understanding the mechanism of MCL-1 ubiquitination is of great importance.

In interphase, MCL-1 is degraded by Mule/ARF-BP1 and SCF β TRCP upon cellular stresses. These ligases do not appear to mediate MCL-1 degradation in mitosis. Instead, previous studies have implicated APC/C-CDC20 (Harley et al.) and SCFFBW7 (Wertz et al.) as E3 ubiquitin ligases for

MCL-1 in mitosis. There were also conflicting reports. For example, contrary to Harley et al., Diaz-Martinez et al. showed that depletion of CDC20 did not cause accumulation of MCL-1 during mitotic arrest. The involvement of SCFFBW7 in mitotic MCL-1 degradation was also questioned by experts in the field. The ubiquitin ligases responsible for MCL-1 degradation in mitosis thus remain to be established.

In this study, Clarke and colleagues (the same group as Harley et al.) provide convincing evidence to rule out SCFFBW7 as the mitotic ubiquitin ligase for MCL-1. They show that APC/C is indeed responsible for MCL-1 mitotic degradation, but this degradation does not require CDC20 (in agreement with Diaz-Martinez et al., but in conflict with Harley et al.). In addition to the D-box that binds at the interface between CDC20 and APC10, MCL-1 contains an IR-tail that interacts with APC3. The D-box is required for MCL-1 degradation, but the IR-tail is not. Instead, the IR tail of MCL-1 appears to inhibit the degradation of other mitotic APC/C substrates. The fact that MCL-1 degradation is dependent on APC/C but not on CDC20 suggests that targeting CDC20 is a better strategy for killing cancer cells in mitosis. APC/C inhibition blocks mitotic exit, but delays MCL-1 degradation and apoptosis onset. CDC20 inhibition blocks mitotic exit without affecting MCL-1 degradation and apoptosis. These findings are highly significant and should be published. However, there are key unanswered questions that need to be addressed experimentally before this paper can be published. This is especially important given the conflicting data on this subject, some of which are from the same authors.

Major points

(1) The same group (Harley et al.) reported previously that CDC20 depletion completely inhibited the mitotic degradation of endogenous MCL-1. By contrast, Diaz-Martinez et al. could not observe accumulation of MCL-1 in CDC20-depleted cells. In the current study, the authors now provide data that are in agreement with Diaz-Martinez et al, but in direct conflict with their own earlier results. This point was not discussed at all. In fact, the Diaz-Martinez paper was not even referenced. Why are their current results different from their previous ones? What changed? This issue needs to be thoroughly discussed and clarified. If their previous results were incorrect, the authors need to have the courage to correct it, instead of trying to brush it under the rug.

(2) The authors provide complicated models and mechanisms to explain APC/C-dependent ubiquitination of MCL-1. Some of the points are complicated and difficult to follow. Because MCL-1 degradation requires the D-box, it has to involve one of the two co-activators of APC/C, CDC20 or CDH1. If CDC20 is not required, it is quite possible that CDH1 is mediating the slow degradation of MCL-1. CDH1 is suppressed by mitotic phosphorylation, but might have basal activities that support gradual degradation of MCL-1. The authors should check whether depletion of CDH1 (alone or together with CDC20 depletion) stabilizes MCL-1 during mitosis.

(3) Because of the challenging nature of identifying the functional ligase for mitotic MCL-1 degradation, it is important for the authors to conclusively prove that APC/C is the ligase. The authors should test whether MCL-1 is ubiquitinated by APC/C-CDC20 or APC/C-CDH1 using reconstituted APC/C assays. MCL-1 Δ D and Δ IR should be included as controls.

Minor points

(1) Western blot analysis should be performed on mitotic cell lysates in the presence or absence of CDC20 to compare the rates of degradation of YFP-MCL-1 and endogenous MCL-1 during mitotic arrest.

(2) In Figure 5C and Extended Figure 3E, the labels for siGAPDH and siCDC20 (or siAPC3) should be reversed.

(3) proTAME binds to APC/C, not to CDC20 (as the authors stated on page 12), and blocks the binding of IR tail of CDC20 to APC/C.

Referee #3:

This study by Allan et al. builds on previous work by the Clarke group uncovering the role of Mcl-1

in directing mitotic cell death. Degradation of Mcl-1 during a delay in mitosis enables cells to induce apoptosis in response to chemotherapeutics activating the spindle checkpoint.

The manuscript provides strong evidence that Mcl-1 is not an Fbw7 substrate in mitosis. Fbw7-mediated Mcl-1 loss thus could not account for mitotic cell killing by paclitaxel, offering a different view than e.g. Wertz et al. (Nature 471, 2011, which ommissively failed to cite Clarke et al.).

Evidence of no role for Fbw7 in mitotic Mcl-1 degradation (Fig. 1) is of significance, although I feel more controls are needed to substantiate the dispute.

Slow degradation of Mcl-1 by APC/C-Cdc20 during a delayed mitosis may act as a sand-clock that, when levels decline below a critical level, permits apoptosis. This however also poses a paradox, as it predicts that apoptosis is induced at normal metaphase, as soon as the spindle checkpoint is satisfied and APC/C-Cdc20 becomes fully active. Previous work showed a role for Mcl-1 phosphorylation by cyclin B1-Cdk1 in fine-tuning the timing of Mcl-1 recognition by the APC/C, which could to some extent resolve this paradox. However, the effect seemed to be partial (Harley et al., EMBO Journal, 2010).

The current study unveils a new mechanism which may explain the paradox better: Mcl-1 is a poor substrate of APC/C-Cdc20 in mitosis, regardless of the spindle checkpoint being active or inactive. This would make a slower sand-clock.

Excitingly, the mechanism that restrains efficient recognition of the D-box region of Mcl-1 by APC/C-Cdc20 appears to require Mcl-1's APC/C interaction motif, the IR tail. Removal of the IR tail renders Mcl-1 a better APC/C-Cdc20 substrate during mitotic arrest, resulting in more rapid degradation of Mcl-1 as soon as the spindle checkpoint is satisfied by Reversine addition (Fig. 4C). Overexpression of the more efficiently cleared IR-tail mutant of Mcl-1 also increases apoptosis in nocodazole-delayed cells (Fig. 5E).

This represents a remarkable and important difference from the effect of a similar APC/C-binding motif in Nek2A, which enhances APC/C-Cdc20-dependent turnover. Furthermore, clarification of the mechanism by which Mcl-1 disappears during a prolonged mitosis may have consequences for the therapeutic response to spindle poisons in Mcl1 (over-)expressing tumors.

Major comments:

1. Figure 1: to compellingly overturn the published concept of Fbw7-dependent Mcl-1 destruction during mitotic arrest, YFP-Mcl-1 degradation should be measured by time-lapse fluorescence microscopy during normal and delayed mitosis in Fbw7 knock-out cells, or a positive control that is clearly stabilized by Fbw7 knock-out or siRNA (Fig 1 D) should be included in the time-lapse experiment.

2. My impression is that some of the results and the main conclusions are difficult to understand for a broader readership, due to the contradictory effects of the mutants and the non-canonical aspects of Mcl-1 as an APC/C substrate. A message as simple as possible (but not simpler) would be required for a paper that also aims to overturn a major concept in the field (that Mcl-1 is an Fbw7 substrate) and support that conclusion with a strong alternative mechanism. The simplest model to which the observations presented lead, I think, is one in which IR-tail pulls Mcl-1 away from the D-box binding site of the APC/C, preventing efficient presentation of the Mcl-1 D-box to Cdc20. This would impair ubiquitination on nearby lysine residues. Removal of the IR tail would restore substrate motility, permitting canonical D-box binding to Cdc20 and checkpoint-dependent destruction. Could the authors comment on this possibility? Ideally, the model would be tested by for instance incorporating a flexible peptide linker before the IR tail, rendering an APC/C substrate which according to this model might be degraded as the Mcl-1 DeltaIR mutant. Regardless, could the authors try to modify the model in Figure 7A to try to illustrate a molecular explanation for their findings? At the moment, Figure 7A illustrates the generic effects on Mcl1-DeltaIR, but preferably the model should focus on the molecular behavior of the normal Mcl1 protein during normal mitosis as compared to a mitotic delay.

3. Is a D-box mutant of Mcl-1-DeltaIR as stable as Mcl1-DeltaIR upon APC11 depletion (Fig. 3B)?

4. Crucially part of the model presented in this study is, secondly, that Mcl-1, even without the IR tail, is a poor Cdc20 substrate, at least in a normal mitosis (e.g. see Fig 4A, disappearance does not

start until after anaphase). This is correctly indicated in the text (p. 7) but should also be emphasized in the abstract and discussion. Could this explain why both YFP-Mcl1 and YFP-Mcl1-DeltaIR are only partially degraded during a normal mitosis (Fig. 4A) or do the authors have a different explanation for these observations?

5. Is the YFP-Mcl1/YFP-Mcl1-DeltaIR degradation from telophase onwards impaired by Cdh1 depletion? (Fig. 4A)? This would further support the claimed APC/C activator-dependence of Mcl-1 degradation.

6. Fig 5C is unclear, the text (p. 11) states: 'prior Cdc20 knockdown restored the usual slow rate of degradation' but the Figure shows accelerated Mcl-1-DeltaIR degradation after Cdc20 depletion. Same confusion for Fig EV3E, and the legend of Fig EV3F is missing.

7. Fig 6A suggests that Mcl-1 has a limited dependency on Cdc20, even though it is a poor substrate, which is intriguing. The same sensitivity to very low Cdc20 levels has been described for the MR-tail APC/C substrate Nek2A and (indeed) suggest a link to the tail (Boekhout and Wolthuis, J Cell Sci, 2015). Is YFP-Mcl1-DeltaIR stabilized by TAME? (This question also relates to the correct interpretation of Fig 5C). And, is YFP-Mcl1 stabilized more effectively by a combination of Cdc20 RNAi and TAME, like Nek2A? This experiment could rule out a role for a second E3 ligase in the residual Mcl-1 degradation seen in Fig. 6A and would substantiate the model that the tail influences the dependency on Cdc20.

Minor points:

1. The authors should discuss a potential additional role of the HUWE ubiquitin ligase in mitotic Mcl-1 degradation (Shi et al., Cancer Res, 71, 2011);
2. Results, p. 6: Nocodazole concentrations should be mentioned in the text. Have the authors tried a mitotic shake-off/chase experiment to analyze Mcl-1 degradation in Fbw7 knock-out cells?
3. p. 7; a citation referring to spindle checkpoint-independent cyclin A and Nek2A degradation is missing.
4. p. 8; 'during mitotic arrest', please define the conditions in the text.
5. Fig. 4. Have the authors tried immunoprecipitating (YFP)-Mcl-1 and Mcl-1DeltaIR to show APC/C -Cdc20 binding during mitotic arrest?
6. The results presented in Fig. 5E suggest a therapeutic strategy of designing Mcl1-IR-tail competitive compounds, predicted to reduce Mcl-1 stability and impairing survival of cancer cells arrested in mitosis. This might be a useful addition to the discussion.

1st Revision - authors' response

21st December 2017

Response to Referees

Allan et al

Atypical APC/C-dependent degradation of Mcl-1 provides an apoptotic timer during mitotic arrest.

Editors requirements

> Please elaborate on the discrepancies reported for the APC/C and Cdc20 contribution in your own earlier work, Wertz et al 2011, Diaz-Martinez et al, and the present study (ref #2 point 1), in order to place your findings adequately in the context of the published literature.

We have corrected the omission of Diaz-Martinez et al (2014) and have discussed this issue in answer to Referee 2 below.

-> Refs #2 (point 2) and ref #3 (point 2) both find that the model for APC/C-dependent ubiquitination of Mcl-1 requires additional experimental data and extensive rewriting/discussion to make the findings more accessible for a non-specialised audience. In our view, this is a valid point that would also go some way to address the concern from ref #1 that the current study remains 'a little undefined in mechanism'

Although the logic of the experiments that we have conducted to examine the interaction between Mcl-1 and APC/C is complex, we have attempted to clarify the manuscript throughout to provide an understanding for the non-specialist. We think that the results are consistent and do provide a clear explanation for the mechanism of interaction between Mcl-1 and APC/C.

-> Regarding in vitro reconstitution of APC/C-Mcl-1 (ref #2) and timelapse movies of Mcl-1 degradation in cells (ref #3) I realize that these may not be trivial experiments and I'd be happy to discuss what kind of data you would be able to include to address these points.

See response to referee 2 below for detailed explanation of attempts to reconstitute the functional interaction between Mcl-1 and APC/C. Unfortunately, we have not been successful despite taking several different approaches. The only in vitro system in which we have had any success has been extracts made from mitotic HeLa cells, which exhibit weak APC/C-dependent degradation that is partially Cdc20 dependent. However, the system has proved to be too inefficient for molecular dissection of the mechanism of Mcl-1 recognition by APC/C and may not properly reproduce the recognition of Mcl-1 during a checkpoint arrest since the checkpoint is not maintained in this in vitro system.

-> Finally, and also related to the previous point, I would ask you to provide a preliminary point-by-point response to all the referee concerns at this stage already. That

should help us define the exact experimental requirements for the revised manuscript. Feel free to contact me with any questions on this.

We thank the referees for their thorough evaluation of our initial manuscript, and for their recognition of its value. Please see point-by-point responses below.

Referee #1:

In this study the authors have investigated the proteolysis of the Mcl1 protein in mitosis. Using FBW7 knockout cells, they find that Mcl1 is targeted by the APC/C and not by SCF-FBW7 as previously reported. This is an important finding as it settles a contentious issue. The authors go on to show that Mcl1 is an unusual APC/C substrate in that it requires little Cdc20 for its degradation and is steadily degraded through mitosis, even when the Spindle assembly checkpoint (SAC) is satisfied. The authors conclude that this behaviour allows Mcl1 degradation to act as a 'mitotic timer' for the apoptotic machinery. The authors show that this behaviour is conferred on Mcl1 by its Isoleucine-Arginine tail that binds to the APC/C. Mcl1 lacking its IR tail is degraded rapidly when the SAC is inactivated. Overall these findings are intriguing - if a little undefined in mechanism - and should be of interest to the readers of The EMBO Journal.

Before publication I think the authors should address the following points:

1) The siRNA experiments of APC8 and Cdc20 are useful to show the difference in sensitivity to the level of the proteins but I would caution the authors against pushing their conclusions too far. From the published structures of the APC/C it is difficult to conceive of an APC/C that lacks APC8; similarly it is difficult to envisage how a protein other than one with a C-box could activate the APC/C in the absence of Cdc20 or Cdh1. My interpretation of these depletion experiments is that the authors have not removed APC8 and Cdc20 completely and that Mcl1 is simply a better substrate because it can bind directly to the APC/C.

The approach we have taken to define the subunit requirements for functional interaction of the APC/C with Mcl-1 are similar to that taken previously by others (e.g. Pines group). While there is of course always the concern that a lack of effect is due to insufficient depletion or that depleting a subunit will disrupt overall structure, we alleviate this concern by always comparing the effect on Mcl-1 degradation with that of the mutant lacking the

IR tail and other, well-documented APC/C substrates. For instance, we show that APC8 depletion *does not* affect the rate of degradation of Mcl-1 or the deltaIR mutant (fig. 3) when it *does* strongly inhibit the loss of cyclin A and cyclin B (fig EV2). Similarly, depletion of Cdc20 (and Cdh1) *does not* prevent the relatively slow rate of Mcl-1 destruction (fig 5A, EV3D) but it *does* inhibit cyclin A destruction (EV3F) and the rapid loss of Mcl-1 deltaIR destruction when the checkpoint is released (Fig. 5C). The rapid, Cdc20-dependent loss of deltaIR Mcl-1 when the checkpoint is released seems to be inconsistent with a model in which the stronger binding of wild-type Mcl-1 to APC/C makes it more sensitive to Cdc20 and yet degraded more slowly. The requirement for a D-box for efficient Mcl-1 turnover by the APC/C even when it is apparently not stimulated by Cdc20 or Cdh1 is unexpected, but it is possible that a role for the D-box in the catalytic turnover and release of the substrate is revealed by a lack of cofactor dependency. Stabilisation of a (non-productive) interaction of Mcl-1 with APC/C by the D box mutation would explain why this mutant has a dominant inhibitory effect of APC/C activity towards other substrates (Fig. 2E).

2) The authors state that TAME binds to Cdc20, but TAME mimics an IR dipeptide and thus binds to the IR-receptor on APC3 (doi.org/10.1016/j.ccr.2010.08.010<<http://doi.org/10.1016/j.ccr.2010.08.010>>). Thus TAME is likely to be in competition with Mcl1 to bind to the APC/C and if Mcl1 affinity for the APC/C is higher this could explain why TAME does not stabilise Mcl1, rather than though the effect of TAME on Cdc20 binding.

We apologise for this inadvertent error of fact, which has been corrected. We agree that it is possible that, on one hand, TAME might not be able to compete with the interaction of Mcl-1 IR tail with the APC/C. Conversely, if TAME interferes with the interaction of the IR tail of Mcl-1 with APC/C, we might expect it to induce more rapid degradation of Mcl-1 like the IR deletion when the checkpoint is released. However, since TAME would presumably also block the Cdc20-dependent degradation of MCL-1 delta IR under these conditions as well as preventing release from mitosis this effect is masked. A specific inhibitor of the interaction of the IR tail of Mcl-1 that does not also block the interaction of the IR tail of Cdc20 would be useful to distinguish between these possibilities but it would probably be difficult to obtain the required selectivity.

Minor point: In Fig 5c have the points been mis-labelled? It appears that Cdc20 knockdown causes faster Mcl1 degradation.

This error has now been corrected

Referee #2:

Mitotic cell death following perturbation of mitosis is failsafe mechanism to prevent the emergence of chromosomal abnormalities. Activation of the spindle checkpoint followed by induction of mitotic cell death by antimetotics is a major strategy in cancer treatment. A critical target of the checkpoint is CDC20, the activator of the anaphase-promoting complex/cyclosome (APC/C), which is a multi-subunit ubiquitin ligase complex that initiates mitotic exit. MCL-1 is an anti-apoptotic BCL-2 family protein that suppresses the activation of caspases. MCL-1 is gradually degraded by ubiquitin-mediated proteolysis during prolonged mitotic arrest. Apoptosis is initiated, when the levels of MCL-1 decline to below the critical threshold. Thus, MCL-1 degradation has been proposed to act a timing device for mitotic cell death. Understanding the mechanism of MCL-1 ubiquitination is of great importance.

In interphase, MCL-1 is degraded by Mule/ARF-BP1 and SCF β TRCP upon cellular stresses. These ligases do not appear to mediate MCL-1 degradation in mitosis. Instead, previous studies have implicated APC/C-CDC20 (Harley et al.) and SCFFBW7 (Wertz et al.) as E3 ubiquitin ligases for MCL-1 in mitosis. There were also conflicting reports. For example, contrary to Harley et al., Diaz-Martinez et al. showed that depletion of CDC20 did not cause accumulation of MCL-1 during mitotic arrest. The involvement of SCFFBW7 in mitotic MCL-1 degradation was also questioned by experts in the field. The ubiquitin ligases responsible for MCL-1 degradation in mitosis thus remain to be established.

In this study, Clarke and colleagues (the same group as Harley et al.) provide convincing evidence to rule out SCFFBW7 as the mitotic ubiquitin ligase for MCL-1. They show that APC/C is indeed responsible for MCL-1 mitotic degradation, but this degradation does not require CDC20 (in agreement with Diaz-Martinez et al., but in conflict with Harley et al.). In addition to the D-box that binds at the interface between CDC20 and APC10, MCL-1 contains an IR-tail that interacts with APC3. The D-box is required for MCL-1 degradation, but the IR-tail is not. Instead, the IR tail of MCL-1 appears to inhibit the degradation of other mitotic APC/C substrates. The fact that MCL-1 degradation is dependent on APC/C but not on CDC20 suggests that targeting CDC20 is a better strategy

for killing cancer cells in mitosis. APC/C inhibition blocks mitotic exit, but delays MCL-1 degradation and apoptosis onset. CDC20 inhibition blocks mitotic exit without affecting MCL-1 degradation and apoptosis. These findings are highly significant and should be published. However, there are key unanswered questions that need to be addressed experimentally before this paper can be published. This is especially important given the conflicting data on this subject, some of which are from the same authors.

Major points

(1) The same group (Harley et al.) reported previously that CDC20 depletion completely inhibited the mitotic degradation of endogenous MCL-1. By contrast, Diaz-Martinez et al. could not observe accumulation of MCL-1 in CDC20-depleted cells. In the current study, the authors now provide data that are in agreement with Diaz-Martinez et al, but in direct conflict with their own earlier results. This point was not discussed at all. In fact, the Diaz-Martinez paper was not even referenced. Why are their current results different from their previous ones? What changed? This issue needs to be thoroughly discussed and clarified. If their previous results were incorrect, the authors need to have the courage to correct it, instead of trying to brush it under the rug.

We apologise for omitting the citation of Diaz-Martinez et al. (2014), a paper that was focussed on other aspects of the control of mitotic cell death but did include data on the lack of effect of Cdc20 depletion on Mcl-1 loss, which we had missed. This reference has now been appropriately included in the Introduction and Discussion. At present, we do not have a perfect explanation for the apparent discrepancy between our previous results (Harley et al, 2010) and our current results, which are in better agreement with Diaz-Martinez et al. (2014) and Sloss et al. (2016) on the lack of requirement for Cdc20 for wild-type Mcl-1 destruction. One possible explanation is that under certain circumstances such as a weak checkpoint arrest, Cdc20 can play a role in Mcl-1 destruction and we have mentioned this possibility in the discussion. However, we have been unable to reproduce our (imperfect) previous experiments in which Cdc20 was only partially depleted and the cells were slipping out of mitosis, indicating a weak checkpoint. We have now greatly improved the efficiency of the Cdc20 knockdown by using a different transfection reagent and protocol that requires a considerably lower concentration of siRNA duplex so it is possible that in the previous experiments we had an off-target effect that stabilised Mcl-1, however unlikely this may seem. However, we have also now established that Cdc20 does play a role in the destruction of Mcl-1 if Mcl-1 does not interact with the APC/C through its C-terminal tail and we are happy to have clarified this role.

(2) The authors provide complicated models and mechanisms to explain APC/C-dependent ubiquitination of MCL-1. Some of the points are complicated and difficult to follow. Because MCL-1 degradation requires the D-box, it has to involve one of the two co-activators of APC/C, CDC20 or CDH1. If CDC20 is not required, it is quite possible that CDH1 is mediating the slow degradation of MCL-1. CDH1 is suppressed by mitotic phosphorylation, but might have basal activities that support gradual degradation of MCL-1. The authors should check whether depletion of CDH1 (alone or together with CDC20 depletion) stabilizes MCL-1 during mitosis.

We have confirmed that Cdh1 is not involved in the mitotic degradation of Mcl-1 (Fig EV3D). We are left with the conclusion that either another, previously recognised cofactor is involved in the D-box dependent recognition of Mcl-1 by APC/C, or that the interaction of Mcl-1 with the APC/C does not require a cofactor and the D-box dependency is due to an effect on the catalytic turnover of the substrate rather than its recruitment. This would explain the dominant inhibitory effect of the D-box mutant on the activity of the APC/C towards other substrates (cyclins A and B, Nek2A; Fig. 2E).

(3) Because of the challenging nature of identifying the functional ligase for mitotic MCL-1 degradation, it is important for the authors to conclusively prove that APC/C is the ligase. The authors should test whether MCL-1 is ubiquitinated by APC/C-CDC20 or APC/C-CDH1 using reconstituted APC/C assays. MCL-1 Δ D and Δ IR should be included as controls.

We assert that the analysis of Mcl-1 degradation in cells, including the subunit dependencies of the APC/C, the effect of mutating the putative D-box and C-terminal IR motifs, and the requirement for APC/C in the ubiquitination of Mcl-1 (Fig. EV3C) can only be explained by Mcl-1 being a direct substrate for APC/C. We have also made extensive attempts to reconstitute in vitro the catalytic interaction of APC/C with Mcl-1 but without success. We have attempted to reproduce the ubiquitination and degradation of Mcl-1 in M-phase *Xenopus* egg extracts (with Dr Hiro Yamano, UCL) and extracts of human HeLa cells arrested in mitosis. Mcl-1 proved to be completely stable in *Xenopus* egg extracts, even though it appeared to be correctly phosphorylated, indicating that some other component is absent from this system. In human cell extracts, we did observe some degradation of in vitro transcribed/translated Mcl-1, but this was much less efficient than mitotic cyclins. The weak degradation was however partially inhibited by TAME indicated that it was Cdc20-APC/C-dependent, suggesting that it was not an ideal model for the usual mechanism of recognition of Mcl-1 by APC/C in mitotically-arrested cells. We have also

attempted to reconstitute the direct interaction between immunoprecipitated APC/C and Mcl-1 in solution, but this was unsuccessful. Possible explanations include the lack of correct mitotic post-translational modification of Mcl-1 and/or APC/C and the absence of an Mcl-1 partner protein such as Noxa; indeed, the correct substrate is unlikely to be monomeric as Mcl-1 associates with other proteins in cells. It is also possible that interaction of Mcl-1 by APC/C is established under checkpoint-inhibited conditions that we are not able to reproduce in vitro. This lack of progress is frustrating, but we think that the cellular experiments provide a clear picture of the functional relationship between Mcl-1 and APC/C and are consistent with Mcl-1 being a direct, albeit unusual, substrate for APC/C.

Minor points

(1) Western blot analysis should be performed on mitotic cell lysates in the presence or absence of CDC20 to compare the rates of degradation of YFP-MCL-1 and endogenous MCL-1 during mitotic arrest.

We have provided data to show that endogenous Mcl-1 and YFP-Mcl-1 are degraded at similar rates during a nocodazole-induced mitotic arrest +/- Cdc20 depletion (Fig. EV3B).

(2) In Figure 5C and Extended Figure 3E, the labels for siGAPDH and siCDC20 (or siAPC3) should be reversed.

We have corrected this mistake.

(3) proTAME binds to APC/C, not to CDC20 (as the authors stated on page 12), and blocks the binding of IR tail of CDC20 to APC/C.

We have corrected this mistake.

Referee #3:

This study by Allan et al. builds on previous work by the Clarke group uncovering the role of Mcl-1 in directing mitotic cell death. Degradation of Mcl-1 during a delay in

mitosis enables cells to induce apoptosis in response to chemotherapeutics activating the spindle checkpoint.

The manuscript provides strong evidence that Mcl-1 is not an Fbw7 substrate in mitosis. Fbw7-mediated Mcl-1 loss thus could not account for mitotic cell killing by paclitaxel, offering a different view than e.g. Wertz et al. (Nature 471, 2011, which ommissively failed to cite Clarke et al.). Evidence of no role for Fbw7 in mitotic Mcl-1 degradation (Fig. 1) is of significance, although I feel more controls are needed to substantiate the dispute.

Slow degradation of Mcl-1 by APC/C-Cdc20 during a delayed mitosis may act as a sand-clock that, when levels decline below a critical level, permits apoptosis. This however also poses a paradox, as it predicts that apoptosis is induced at normal metaphase, as soon as the spindle checkpoint is satisfied and APC/C-Cdc20 becomes fully active. Previous work showed a role for Mcl-1 phosphorylation by cyclin B1-Cdk1 in fine-tuning the timing of Mcl-1 recognition by the APC/C, which could to some extent resolve this paradox. However, the effect seemed to be partial (Harley et al., EMBO Journal, 2010).

The current study unveils a new mechanism which may explain the paradox better: Mcl-1 is a poor substrate of APC/C-Cdc20 in mitosis, regardless of the spindle checkpoint being active or inactive. This would make a slower sand-clock.

Excitingly, the mechanism that restrains efficient recognition of the D-box region of Mcl-1 by APC/C-Cdc20 appears to require Mcl-1's APC/C interaction motif, the IR tail. Removal of the IR tail renders Mcl-1 a better APC/C-Cdc20 substrate during mitotic arrest, resulting in more rapid degradation of Mcl-1 as soon as the spindle checkpoint is satisfied by Reversine addition (Fig. 4C). Overexpression of the more efficiently cleared IR-tail mutant of Mcl-1 also increases apoptosis in nocodazole-delayed cells (Fig. 5E).

This represents a remarkable and important difference from the effect of a similar APC/C-binding motif in Nek2A, which enhances APC/C-Cdc20-dependent turnover. Furthermore, clarification of the mechanism by which Mcl-1 disappears during a prolonged mitosis may have consequences for the therapeutic response to spindle poisons in Mcl1 (over-)expressing tumors.

Major comments:

1. Figure 1: to compellingly overturn the published concept of Fbw7-dependent Mcl-1 destruction during mitotic arrest, YFP-Mcl-1 degradation should be measured by time-lapse fluorescence microscopy during normal and delayed mitosis in Fbw7 knock-out cells, or a positive control that is clearly stabilized by Fbw7 knock-out or siRNA (Fig 1 D) should be included in the time-lapse experiment.

In Fig. EV1F, we now show that the depletion of Fbw7 in YFP-Mcl-1 expressing cells is sufficient to stabilise Cyclin E and Myc, both well-established targets for SCF^{Fbw7} during interphase.

We also attempted to analyse YFP-Mcl-1 degradation in DLD-1 and HCT116 cells in which Fbw7 was deleted. However, we were unable to analyse this process by live-cell microscopy due to technical difficulties:

- (i) We tried both transiently-transfected cells and made stably-expressing cells but in both cases the expression levels of YFP-Mcl-1 were far too high for live-cell analysis. Reduction of the amount of DNA used in the transient transfection resulted in only fewer cells expressing YFP-Mcl-1 rather than lower levels of expression.
- (ii) Neither DLD-1 nor HCT116 cell lines were viable for long in L15 CO₂-free media required for live-cell imaging.

Concerning a positive control for the activity of SCF^{Fbw7} during mitosis, we are not aware of any such characterised substrate.

2. My impression is that some of the results and the main conclusions are difficult to understand for a broader readership, due to the contradictory effects of the mutants and the non-canonical aspects of Mcl-1 as an APC/C substrate. A message as simple as possible (but not simpler) would be required for a paper that also aims to overturn a major concept in the field (that Mcl-1 is an Fbw7 substrate) and support that conclusion with a strong alternative mechanism. The simplest model to which the observations presented lead, I think, is one in which IR-tail pulls Mcl-1 away from the D-box binding site of the APC/C, preventing efficient presentation of the Mcl-1 D-box to Cdc20. This would impair ubiquitination on nearby lysine residues. Removal of the IR tail would restore substrate motility, permitting canonical D-box binding to Cdc20 and checkpoint-dependent destruction. Could the authors comment on this possibility? Ideally, the model would be tested by for instance incorporating a flexible peptide linker before the IR tail, rendering an APC/C substrate which according to this model might be degraded as the Mcl-1 DeltaIR mutant. Regardless, could the authors try to modify the model in Figure 7A to try to illustrate a molecular explanation for their findings? At the moment, Figure 7A illustrates the generic effects on Mcl1-DeltaIR, but preferably the model should focus on the molecular behaviour of the normal Mcl1 protein during normal mitosis as compared to a mitotic delay.

We have attempted to simplify the text describing the experiments. We disagree that the mutants have contradictory effects although we concede that the proposed mechanism is non-canonical (and therefore interesting).

Concerning the previous assertion (Wertz et al, 2011) that Mcl-1 is a substrate for SCF^{Fbw7} during mitosis, this finding was based on a biochemical analysis of a mixed population of mitotic and interphase cells. We have shown that deletion of Fbw7 does indeed cause cells to exit mitosis more readily from an arrest caused by microtubule poisons (Fig. EV1B) which could explain the apparent effects on Mcl-1 stability and resistance to paclitaxel when Fbw7 is lost. Note that we have used the same FBW7 knockout cell lines for this analysis as those used by Wertz et al. Under conditions where we demonstrate that slippage in FBW7 knockout cells was reduced considerably (increased taxol concentration), Mcl-1 degradation was unaffected by FBW7 status. It remains possible however, that Mcl-1 is a substrate for SCF^{Fbw7}, as well as Mule/ARF-BP1/HUWE and SCF^{βTrCP}, during interphase. We were more confident of our earlier assertion (Harley et al, 2010) that the degradation of Mcl-1 in mitotic cells was dependent on APC/C because of the different protocol used in which floating mitotic cells were separated from adherent interphase cells. To conclusively demonstrate a role during mitosis in this study, we have used live-cell imaging in which we can be certain that the degradation is occurring during mitosis. This approach also provides a minute-by-minute analysis of rates, which is impossible using the biochemical approach.

The details of the molecular mechanism by which Mcl-1 interacts with the APC/C are suggested by the cellular analysis we have carried out, but confirmation would require a molecular structure and the dynamics of the process would still have to be inferred. In essence, however, we agree in principle that the data support the mechanism proposed by the referee, i.e. that the IR tail restricts the efficient ubiquitination of Mcl-1 by APC/C and prevents stimulation by Cdc20. However, it is only speculative at present how Mcl-1 would be orientated on the APC/C complex. A caveat to this model is the inhibitory effect of mutating the D-box on Cdc20-independent Mcl-1 degradation (and therefore presumably recognition of Mcl-1 by the APC/C), so it does not seem to be quite that simple.

We have attempted to draw a molecular model for the interaction of Mcl-1 with the APC/C from the known APC/C structures, but because of the complexity of the projection and the absence of a complete structure for Mcl-1 that includes the IR tail, we decided it was not helpful.

3. Is a D-box mutant of Mcl-1-DeltaIR as stable as Mcl1-DeltaIR upon APC11 depletion (Fig. 3B)?

We did not understand the usefulness of this experiment other than to try another combination of mutations.

4. Crucially part of the model presented in this study is, secondly, that Mcl-1, even without the IR tail, is a poor Cdc20 substrate, at least in a normal mitosis (e.g. see Fig 4A, disappearance does not start until after anaphase). This is correctly indicated in the text (p. 7) but should also be emphasized in the abstract and discussion. Could this explain why both YFP-Mcl1 and YFP- Mcl-1-DeltaIR are only partially degraded during a normal mitosis (Fig. 4A) or do the authors have a different explanation for these observations?

We agree that Mcl-1 is, compared to cyclin A during mitosis and cyclin B after release of the checkpoint, a poor APC/C substrate, and we have tried to further emphasise this in the abstract and discussion. That Mcl-1 is not efficiently removed in a normal mitosis is crucial, we propose, to its action, so that it protects against apoptosis during an unperturbed mitosis but allowing apoptosis after a prolonged arrest. This function is illustrated by the data shown in Fig. 6C.

5. Is the YFP-Mcl-1/YFP-Mcl1-DeltaIR degradation from telophase onwards impaired by Cdh1 depletion? (Fig. 4A)? This would further support the claimed APC/C activator-dependence of Mcl-1 degradation.

From our experiments using cells progressing through mitosis (Fig. 4A), it looks as if YFP-Mcl-1 is stabilised around telophase; this may also involve post-translational changes such as dephosphorylation. The relative instability of the deltaIR mutant at this stage of mitosis might conceivably involve Cdh1, but experiments in which release from the checkpoint is triggered by reversine (Fig. EV3H), conditions that may be similar to normal telophase, show that degradation of Mcl-1-deltaIR is rather Cdc20-dependent under these conditions.

6. Fig 5C is unclear, the text (p. 11) states: 'prior Cdc20 knockdown restored the usual slow rate of degradation' but the Figure shows accelerated Mcl-1-DeltaIR degradation after Cdc20 depletion. Same confusion for Fig EV3E, and the legend of Fig EV3F is missing.

We apologise for this confusion and we have now corrected the labels.

7. Fig 6A suggests that Mcl-1 has a limited dependency on Cdc20, even though it is a poor substrate, which is intriguing. The same sensitivity to very low Cdc20 levels has been described for the MR-tail APC/C substrate Nek2A and (indeed) suggest a link to the tail (Boekhout and Wolthuis, J Cell Sci, 2015). Is YFP-Mcl1-DeltaIR stabilized by TAME? (This question also relates to the correct interpretation of Fig 5C). And, is YFP-Mcl1 stabilized more effectively by a combination of Cdc20 RNAi and TAME, like Nek2A? This experiment could rule out a role for a second E3 ligase in the residual Mcl-1 degradation seen in Fig. 6A and would substantiate the model that the tail influences the dependency on Cdc20.

Fig. 6A does not support a role for Cdc20 in the degradation of Mcl-1. However, although possibly partial effects of inhibitors in combinations with mutations can be over-interpreted, we have not yet found a condition in which Mcl-1 destruction is blocked as efficiently as the degradation of cyclin B by TAME or depletion of APC2/11 (Fig. 6B). It remains possible, therefore, that another pathway contributes to the destruction of Mcl-1 during mitotic arrest (one not dependent on Fbw7).

Minor points:

1. The authors should discuss a potential additional role of the HUWE ubiquitin ligase in mitotic Mcl-1 degradation (Shi et al., Cancer Res, 71, 2011);

We have not analysed a potential role for Mule/ARF-BP1/HUWE in the mitotic degradation of Mcl-1 in this study so this remains an open possibility. It could however only contribute a basal activity compared to APC/C.

2. Results, p. 6: Nocodazole concentrations should be mentioned in the text. Have the authors tried a mitotic shake-off/chase experiment to analyze Mcl-1 degradation in Fbw7 knock-out cells?

We have noted the concentration of nocodazole used (830nM except where stated) in the Results as well as in the Methods and Figure Legends. We did try to analyse Mcl-1 degradation in Fbw7 knockout cells but we found that the DLD-1 and HCT116 cells (WT and Fbw7 KO) were not adherent enough to obtain a pure enough population to analyse or replat for further timepoints after rerelease from mitosis. This is why we carried out a timecourse following thymidine block and then release into a sufficient concentration of paclitaxel (500 nM) to maintain a mitotic arrest of several hours (Figs 1B, EV1B, EV1C).

3. p. 7; a citation referring to spindle checkpoint-independent cyclin A and Nek2A degradation is missing.

We have cited Hayes et al. (2008) in this context, which refers particularly to Nek2A degradation.

4. p. 8; 'during mitotic arrest', please define the conditions in the text.

Routinely, this refers to arrest induced by a high concentration of nocodazole (830 nM) as described.

5. Fig. 4. Have the authors tried immunoprecipitating (YFP)-Mcl-1 and Mcl-1DeltaIR to show APC/C -Cdc20 binding during mitotic arrest?

We have indeed attempted to observe an interaction between Mcl-1 and APC during mitotic arrest. However, we have been unsuccessful in obtaining consistent results, which might be due to the transient nature of their interaction.

6. The results presented in Fig. 5E suggest a therapeutic strategy of designing Mcl1-IR-tail competitive compounds, predicted to reduce Mcl-1 stability and impairing survival of cancer cells arrested in mitosis. This might be a useful addition to the discussion.

We have added a paragraph at the end of the Discussion to include this interesting suggestion.

2nd Editorial Decision

9th March 2018

Thank you for submitting a revised version of your manuscript and my apologies for the extended duration of the review period. Your study has now been seen by all three original referees and their comments are shown below.

As you will see, referees #1 and #2 are overall satisfied with the revised version and support publication following minor text revisions. Ref #3 is more critical and finds that several of the original issues raised have not been sufficiently clarified. However, it is also clear that most of ref #3's concerns can be addressed via adjustments of the text and more extensive (and careful) discussions of the existing literature and the conclusions that can be drawn from the data.

In conclusion, the impression from the referees is that while your study may not fully solve the question of Mcl-1 degradation in mitosis it provides interesting data on this important topic and should therefore be published in The EMBO Journal once the remaining issues have been adequately incorporated. From my side, I think this can be done with textual changes only, although if you happen to have data on the Mcl-1 double mutant referred to by ref #3 I would suggest you to include it in the final version of the manuscript. I would therefore invite you to submit such a final revision using the link provided below. In addition to addressing the remaining concerns from the

referees, please also include the following editorial points:

-> Please fill out and include an author checklist as listed in our online guidelines (<http://emboj.embopress.org/authorguide>)

-> When referring to the Expanded View figures in the text, please use the nomenclature fig EV1, fig EV2 etc. We also noticed that there is currently no call-out to figure EV4A.

-> Please include scale bars for the microscopy images in figs 1-4

-> We generally encourage the publication of source data, particularly for electrophoretic gels and blots, with the aim of making primary data more accessible and transparent to the reader. We would need 1 file per figure (which can be a composite of source data from several panels) in jpg, gif or PDF format, uploaded as "Source data files". The gels should be labelled with the appropriate figure/panel number, and should have molecular weight markers; further annotation would clearly be useful but is not essential. These files will be published online with the article as a supplementary "Source Data". Please let me know if you have any questions about this policy.

Thank you again for giving us the chance to consider your manuscript for The EMBO Journal, I look forward to receiving your final revision.

REFeree REPORTS

Referee #1:

In their revised study the authors have addressed my concerns: I am still of the view that the uncertainties of siRNA preclude concluding that any protein is not required for a process. To make these conclusions a genetic knockout is required, as the authors themselves have shown with the FBW7 knockout cell line. I suspect that Mcl1 will eventually be shown to require Cdc20 (or another cofactor); nevertheless, the demonstration that Mcl1 is likely to be an APC/C substrate and is not a FBW7 substrate is important and warrants publication.

One minor point: the authors now reference Hayes et al 2008 as a reference for cyclin A and Nek2A destruction. This is not appropriate - these 2 proteins are degraded by completely different mechanisms, If the authors wish to include Cyclin A as a SAC-independent substrate they should reference Di Fiore and Pines, 2010 DOI: 10.1083/jcb.201001083.

Referee #2:

The authors have made a good effort to address my concerns. They could not demonstrate the direct ubiquitination of MCL1 by APC/C in vitro. They did not mention whether they attempted to reconstitute the ubiquitination of MCL1^{ΔIR}, which should be a conventional APC/C-Cdc20 substrate. If they did and the results are negative, they should explicitly state that. With this said, given the large amount of data supporting a requirement of APC/C in MCL1 degradation in human cells, I think that the main conclusion of the paper is adequately supported. I support the publication of this excellent study.

Referee #3:

I wish to thank the authors for the many additional experiments attempted in response to the reviewers comments to address the important -and clearly cancer-relevant- mechanisms underlying Mcl-1 protein stability and homeostasis during mitosis.

The results have, to some extent regrettably, not fully solved the -in my view- paradoxical roles of the IR tail and D-box in Mcl-1's degradation. Nevertheless, the evidence that Mcl-1 is an unusual APC/C substrate (and the APC/C is as yet poorly understood E3 ligase for it) is highly

important.

I have several final remarks summarized in three points, which I will leave to the editor for further evaluation. I do recommend a further discussion of these points prior to publication.

Remaining points

1. The title doesn't fully cover the exact news value of the current study. Mcl-1 has already been known to be an apoptotic timer during mitotic arrest for a while. The novelty here is mainly the non-canonical role for the APC/C in the process (e.g. in which engagement of Mcl-1 to the APC/C might be negatively affecting its recognition as a substrate).

So, the current study in particular reveals roles of the APC/C in controlling levels of a crucial mitotic survival factor, rather than uncovering a pro-survival effect of high mitotic Mcl-1 expression levels per se. The title could reflect this more adequately I think.

2. Discussion of literature

Page 3. The introduction is slightly out of date. Regarding Mcl-1, a debate the reader needs to be acknowledged of relates to whether Mcl-1 is substrate to an E3 ubiquitin ligase at all or might be degraded by the proteasome independently of a requirement for ubiquitination (Sloss et al., 2016). Furthermore, Mcl-1, as well as its upstream regulators, might also influence the rate of cyclin B1 destruction and duration of the mitotic arrest (Sloss et al., 2016; Diaz-Martinez, 2014), as a matter of network interference. This needs to be introduced in the Introduction section or discussed (e.g. in relation to Figure 2E).

3. Model

My revised view on the basis of the comments in the rebuttal is that the MR tail 'hides' Mcl-1 'at the back of its murderer', in which APC3 seems to be the 'back' in line with earlier reports of it being a receptor subunit for APC/C substrates' IR tails. This would render the degradation of (wt) Mcl-1 largely Cdc20 independent. I think Figure 5C shows that in the absence of a possibility to 'hide' and in the presence of a fully active APC/C, Cdc20 helps to attack Mcl-1 (missing here is the Mcl-1 double mutant: without D box and without IR tail). Figure 5D suggests again that, when unable to 'hide', Mcl-1 is degraded more efficiently by any residual APC/C-Cdc20 activity.

Regarding Figure 6, and remark 2 of Reviewer 1, I think caution is needed. There is no evidence that TAME prohibits Mcl-1 binding to the APC/C, while there is clear evidence that TAME predominantly inhibits Cdc20 and arrests cells in the spindle checkpoint (Zhen et al, <https://doi.org/10.1016/j.ccr.2010.08.010>). This could also explain why the TAME degradation curves of Mcl-1 in Figure 6A (in TAME) look similar to those in Figure 5 (in the spindle checkpoint).

In discussing the incomplete stabilization of Mcl-1 even by the combination of APC2 and APC11 siRNA (Figure 6A) or D-box mutation (Figure 2B), the authors should refer to the observation that a lysine-less Mcl-1 mutant behaves as a wt Mcl-1 in terms of protecting cells against apoptosis during mitotic delay, suggesting that a ubiquitin-independent degradation route also contributes to Mcl-1 turn-over (Sloss et al, 2016). (A proper positive control here would have been to show full stabilization of YFP-Mcl-1 by proteasome inhibition in the used setting).

Corrections

Fig 3E. Lanes are not aligned vertically making the figure hard to read.

Fig 3, Fig 4A. Mcl-1DIR is misspelled (Delta IR).

Fig 6B right panel siACP2 -> siAPC2

Response to Referees (version 2)

Allan et al

Atypical APC/C-dependent degradation of Mcl-1 provides an apoptotic timer during mitotic arrest.

Editorial points:

-> Please fill out and include an author checklist as listed in our online guidelines (<http://emboj.embopress.org/authorguide>)

We have provided a completed checklist.

-> When referring to the Expanded View figures in the text, please use the nomenclature fig EV1, fig EV2 etc. We also noticed that there is currently no call-out to figure EV4A.

We have made these corrections

-> Please include scale bars for the microscopy images in figs 1-4

Scale bars have been added.

-> We generally encourage the publication of source data, particularly for electrophoretic gels and blots, with the aim of making primary data more accessible and transparent to the reader. We would need 1 file per figure (which can be a composite of source data from several panels) in jpg, gif or PDF format, uploaded as "Source data files". The gels should be labelled with the appropriate figure/panel number, and should have molecular weight markers; further annotation would clearly be useful but is not essential. These files will be published online with the article as a supplementary "Source Data". Please let me know if you have any questions about this policy.

Source data of complete western blots has been provided.

Referee #1:

In their revised study the authors have addressed my concerns: I am still of the view that the uncertainties of siRNA preclude concluding that any protein is not required for a process. To make these conclusions a genetic knockout is required, as the authors themselves have shown with the FBW7 knockout cell line. I suspect that Mcl1 will eventually be shown to require Cdc20 (or another cofactor); nevertheless, the demonstration that Mcl1 is likely to be an APC/C substrate and is not a FBW7 substrate is important and warrants publication.

One minor point: the authors now reference Hayes et al 2008 as a reference for cyclin A and Nek2A destruction. This is not appropriate - these 2 proteins are degraded by completely different mechanisms, If the authors wish to include Cyclin A as a SAC-independent substrate they should reference Di Fiore and Pines, 2010 DOI: 10.1083/jcb.201001083.

This reference has been added.

Referee #2:

The authors have made a good effort to address my concerns. They could not demonstrate

the direct ubiquitination of MCL1 by APC/C in vitro. They did not mention whether they attempted to reconstitute the ubiquitination of MCL1 deltaIR, which should be a conventional APC/C-Cdc20 substrate. If they did and the results are negative, they should explicitly state that. With this said, given the large amount of data supporting a requirement of APC/C in MCL1 degradation in human cells, I think that the main conclusion of the paper is adequately supported. I support the publication of this excellent study.

Referee #3:

I wish to thank the authors for the many additional experiments attempted in response to the reviewers comments to address the important -and clearly cancer-relevant- mechanisms underlying Mcl-1 protein stability and homeostasis during mitosis.

The results have, to some extent regrettably, not fully solved the -in my view- paradoxical roles of the IR tail and D-box in Mcl-1's degradation. Nevertheless, the evidence that Mcl-1 is an unusual APC/C substrate (and the APC/C is an as yet poorly understood E3 ligase for it) is highly important.

I have several final remarks summarized in three points, which I will leave to the editor for further evaluation. I do recommend a further discussion of these points prior to publication.

Remaining points

1. The title doesn't fully cover the exact news value of the current study. Mcl-1 has already been known to be an apoptotic timer during mitotic arrest for a while. The novelty here is mainly the non-canonical role for the APC/C in the process (e.g. in which engagement of Mcl-1 to the APC/C might be negatively affecting its recognition as a substrate). So, the current study in particular reveals roles of the APC/C in controlling levels of a crucial mitotic survival factor, rather than uncovering a pro-survival effect of high mitotic Mcl-1 expression levels per se. The title could reflect this more adequately I think.

We take on board the referee's comment that it is already understood that Mcl-1 protects cells in mitosis but we think the existing title emphasises that it is the timing aspect of its destruction and that this is determined by an unusual APC-C dependent mechanism accurately reflects the key findings of the paper.

2. Discussion of literature

Page 3. The introduction is slightly out of date. Regarding Mcl-1, a debate the reader needs to be acknowledged of relates to whether Mcl-1 is substrate to an E3 ubiquitin ligase at all or might be degraded by the proteasome independently of a requirement for ubiquitination (Sloss et al., 2016). Furthermore, Mcl-1, as well as its upstream regulators, might also influence the rate of cyclin B1 destruction and duration of the mitotic arrest (Sloss et al., 2016; Diaz-Martinez, 2014), as a matter of network interference. This needs to be introduced in the Introduction section or discussed (e.g. in relation to Figure 2E) .

Citations to these papers have now been repeated in the initial paragraph of the Discussion where their relevance is highlighted.

3. Model

My revised view on the basis of the comments in the rebuttal is that the MR tail 'hides' Mcl-1 'at the back of its murderer', in which APC3 seems to be the 'back' in line with earlier reports of it being a receptor subunit for APC/C substrates' IR tails. This would render the degradation of (wt) Mcl-1 largely Cdc20 independent. I think Figure 5C shows that in the absence of a possibility to 'hide' and in the presence of a fully active APC/C, Cdc20 helps to attack Mcl-1 (missing here is the Mcl-1 double mutant: without D box and

without IR tail). Figure 5D suggests again that, when unable to 'hide', Mcl-1 is degraded more efficiently by any residual APC/C-Cdc20 activity.

Regarding Figure 6, and remark 2 of Reviewer 1, I think caution is needed. There is no evidence that TAME prohibits Mcl-1 binding to the APC/C, while there is clear evidence that TAME predominantly inhibits Cdc20 and arrests cells in the spindle checkpoint (Zhen et al, <https://doi.org/10.1016/j.ccr.2010.08.010>). This could also explain why the TAME degradation curves of Mcl-1 in Figure 6A (in TAME) look similar to those in Figure 5 (in the spindle checkpoint).

We thank the referee for these thoughts but we think that further speculation about the mechanism of Mcl-1's recognition by the APC/C is not warranted in the paper.

In discussing the incomplete stabilization of Mcl-1 even by the combination of APC2 and APC11 siRNA (Figure 6A) or D-box mutation (Figure 2B), the authors should refer to the observation that a lysine-less Mcl-1 mutant behaves as a wt Mcl-1 in terms of protecting cells against apoptosis during mitotic delay, suggesting that a ubiquitin-independent degradation route also contributes to Mcl-1 turn-over (Sloss et al, 2016). (A proper positive control here would have been to show full stabilization of YFP-Mcl-1 by proteasome inhibition in the used setting).

This observation of Sloss et al has now been mentioned in the initial paragraph of the Discussion.

Corrections

Fig 3E. Lanes are not aligned vertically making the figure hard to read.

Fig 3, Fig 4A. Mcl-1DIR is misspelled (Delta IR).

Fig 6B right panel siACP2 -> siAPC2

These corrections have been made.

Corresponding Author Name: Paul Clarke

EMBO Journal

Manuscript Number: EMBOJ-2017-96831